# Steller's sea cow genome suggests this species began going extinct before the arrival of Paleolithic humans

Fedor S. Sharko[1,2,8], Eugenia S. Boulygina[1,8], Svetlana V. Tsygankova[1], Natalia V. Slobodova[1], Dmitry A. Alekseev[3], Anna A. Krasivskaya[4], Sergey M. Rastorguev [1], Alexei N. Tikhonov[5,6] & Artem V. Nedoluzhko [7✉]

Anthropogenic activity is the top factor directly related to the extinction of several animal species. The last Steller's sea cow (*Hydrodamalis gigas*) population on the Commander Islands (Russia) was wiped out in the second half of the 18th century due to sailors and fur traders hunting it for the meat and fat. However, new data suggests that the extinction process of this species began much earlier. Here, we present a nuclear de novo assembled genome of *H. gigas* with a 25.4× depth coverage. Our results demonstrate that the heterozygosity of the last population of this animal is low and comparable to the last woolly mammoth population that inhabited Wrangel Island 4000 years ago. Besides, as a matter of consideration, our findings also demonstrate that the extinction of this marine mammal starts along the North Pacific coastal line much earlier than the first Paleolithic humans arrived in the Bering sea region.

[1] National Research Center "Kurchatov Institute", 1st Akademika Kurchatova Square, Moscow, Russia. [2] Research Center of Biotechnology of the Russian Academy of Sciences, Moscow, Russia. [3] Russian Presidential Academy of National Economy and Public Administration, Prospect Vernadskogo, 82, Moscow, Russia. [4] Skolkovo Institute of Science and Technology, Moscow, Russia. [5] Zoological Institute Russian Academy of Sciences, Universitetskaya nab., 1, Saint-Petersburg, Russia. [6] Institute of Applied Ecology of the North, North-Eastern Federal University, Yakutsk, Russia. [7] Faculty of Biosciences and Aquaculture, Nord University, Bodø, Norway. [8] These authors contributed equally: Fedor S. Sharko, Eugenia S. Boulygina. ✉email: artem.nedoluzhko@nord.no

The Steller's sea cow (*Hydrodamalis gigas*) is an extinct species from Sirenia order (Fig. 1A), which inhabited the coastal areas of the North Pacific Ocean (including the Bering Sea) during the Pleistocene and Holocene (Fig. 1B). The Bering sea level significantly dropped during the Late Pleistocene glaciations[1], which likely provided new coastal habitats for Steller's sea cow and other sirenian species. The *H. gigas* population thrived during this period, and its distribution range was extended to the entire North Pacific, from Japan through the Aleutian Islands to California[2]. Interestingly, another sirenian species—dugongs—also inhabited extensive areas in the South Pacific Ocean coastlines (up to Tasmania), and manatees lived along Atlantic Ocean from South America to Ohio River on the north[3–5]. Then, at the transition from the Late Pleistocene to Early Holocene, according to paleontological studies, the sirenian distribution area significantly fragmented[6], and this was possibly caused by the increasing global temperatures and sea-level rise[7].

The Sirenia order currently contains only four species from two genera, the dugong (*Dugong dugon*) and manatees (West Indian manatee—*Trichechus manatus*, Amazonian manatee—*T. inunguis*, and African manatee—*T. senegalensis*), that inhabit the warm waters of the Atlantic, Pacific, and Indian oceans. The dugong is the closest species to *H. gigas* as has been shown in previous studies[8,9].

The last population of Steller's sea cow was discovered by the Vitus Bering's Great Northern Expedition (1741) along the coasts of the Commander Islands (Medny and Bering Islands), and it was gone by the end of the 18th century (Fig.1A).

*H. gigas* was up to 11 tons in mass and up to 10 meters in length; however, the adult animals from the last population that was discovered on the Commander Islands were significantly smaller. Biologist Georg Wilhelm Steller (1709–1746), who participated in the Vitus Bering's Expedition, stated that these sirenian species weighted around 4500–5900 kg and had a body length of ~750 cm. He also documented this animal as a stenobiontic species that adapted to brown algae diets (mainly kelp) and the cold coastal conditions of the Bering Sea region[6].

Recent studies supposed that the climate changes and Paleolithic human hunting may have reduced *H. gigas* population levels before European sailors made the last strike in the evolutionary history of this species[2,10,11]. Researchers also supposed that Steller's sea cow was doomed to extinction, even without the direct killing of the last population, because of the overhunting on sea otters and the co-occurring loss of kelp forests in this region[12].

Here, we present a historical Steller's sea cow nuclear de novo assembled genome (25.4-fold) from Commander Islands (Russia) and use it to investigate the population history of this marine

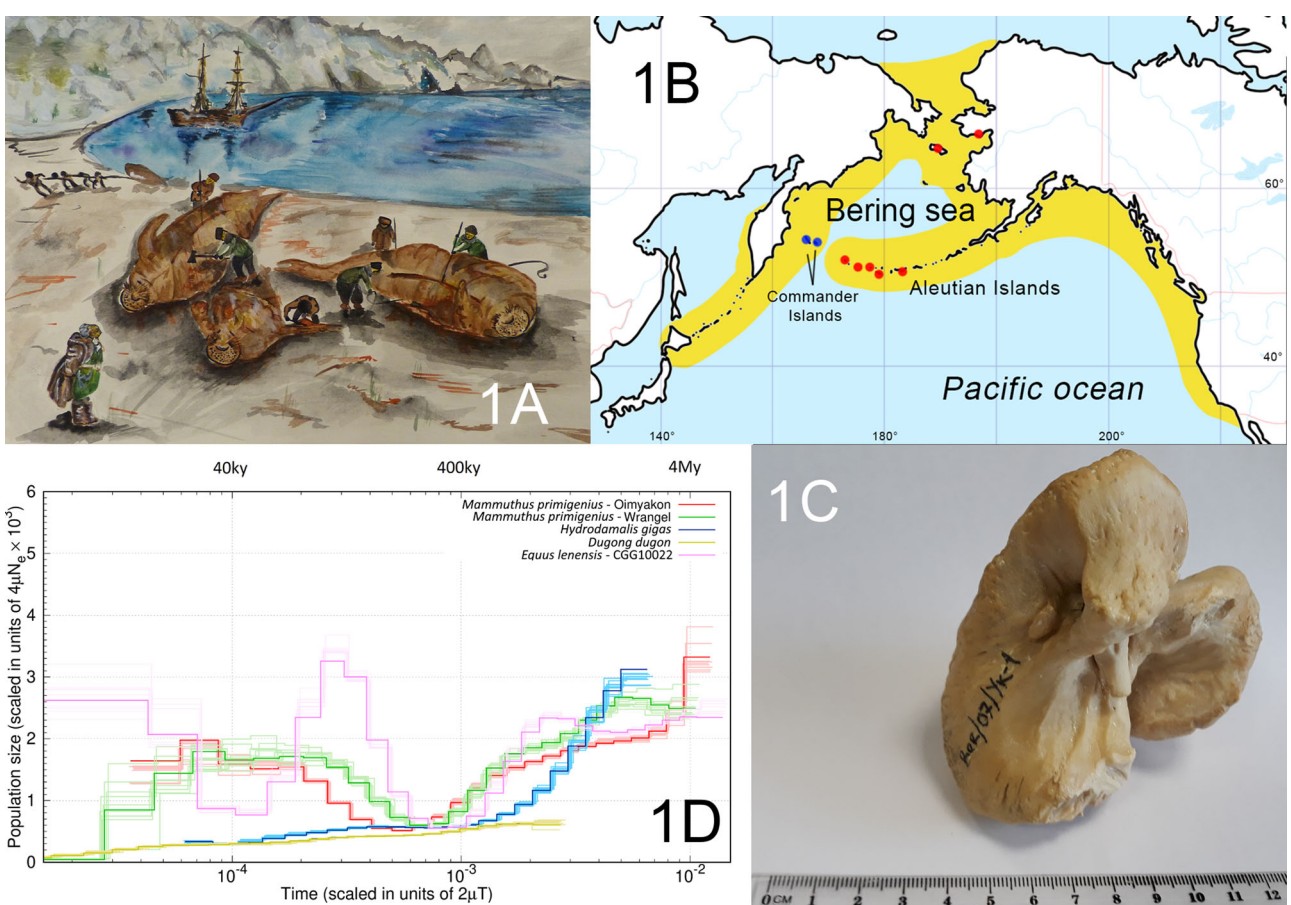

**Fig. 1 Genome sequencing of the extinct Steller's sea cow. A** Reconstruction, aquarelle: The sailors of the Vitus Bering's Great Northern Expedition (1741) cutting up the carcasses of killed Steller's sea cows. Artist: Ekaterina Khritonenkova. **B** Geographical map of the Bering Sea region showing the distribution range of Steller's sea cow during the Late Pleistocene (yellow), the archeological sites and localities (red circles) where samples of Steller's sea cow were described, and the distribution of the last *H. gigas* population (blue circles). **C** The *Hydrodamalis gigas* petrous bone that was used for historical DNA extraction. **D** Steller's sea cow, dugong, woolly mammoths, and Lena horse population size histories inferred using the Pairwise Sequentially Markovian Coalescent (PSMC) model. Population size history of each species marked by different color. Time is given in units of divergence per base pair on the X-axis, effective population size is shown on the Y-axis. Ky—means thousand years ago. My—means million years ago.

mammal during the last several millions of years in the Bering Sea region. It is intriguing that the demographic trajectories of an extinct terrestrial woolly mammoth and Pleistocene Lena horse, as well as modern marine species, differ from the sirenians, including the extinct Steller's sea cow and extant dugong. Terrestrial animals (woolly mammoth and Lena horse) have no less than two population bottlenecks during the Middle or Early Pleistocene while the *H. gigas* and *D. dugon* populations have only one catastrophic population decline, and it bottomed out around 400 thousand years ago. *H. gigas* and *D. dugon* show the same demographic trajectory even in comparison with modern Cetacea and Pinnipedia, such as narwhal or walrus. Steller's sea cow shared several genes evolved under positive selection with these species. A part of these genes are related to metabolic and immune functions, and the most interesting of them is *leptin*, which participates in energy homeostasis and reproductive regulation in modern marine mammals[13]. This finding demonstrates convergent evolution among modern marine species as well as extinct species for several reasons. Our results clearly show that Steller's sea cow definitively embarked on the road toward extinction long before the arrival of the first Paleolithic hunter-gatherers in the Beringia, which is estimated to have occurred at least 25,000–30,000 years ago[14,15].

## Results

**DNA sequencing statistics.** The *H. gigas* museum specimen that is housed in the Museum of the Ocean (Kaliningrad, Russia) was used for the sampling of bone powder from a petrous bone. We conducted several extractions that were used for library preparation and DNA sequencing. We needed to select libraries with relatively high endogenous DNA content, and so the test-sequencing run (paired-end, 2 × 150 bp) was used the first time (Supplementary Table 1). The total number of reads generated for six *H. gigas* DNA-libraries varied from 6,531,468 to 7,810,724 per DNA-library. A number of endogenous reads was measured for each DNA-library with a PALEOMIX v1.2.14 pipeline[16].

Four DNA-libraries (Lib1k, Lib3k, Lib4-2, and St11) with a high percentage of "historical" DNA reads (>40%) were used for the deep sequencing (paired-end, 2 × 50 bp). The total number of reads generated for these four *H. gigas* DNA-libraries varied from 412,855,578 to 480,168,784 per DNA-library, half of which could be uniquely mapped to the West Indian Manatee—*T. manatus* reference genome sequence ("TriManLat1.0 [https://www.ncbi.nlm.nih.gov/assembly/GCF_000243295.1/]") (Supplementary Table 2).

**DNA preservation of Steller's sea cow specimen.** We showed that well-preserved Steller's sea cow petrous bone (Fig. 1C) was a good source of endogenous DNA (~55%) (Supplementary Table 2). The usage of this type of cranium bone helped us to generate 25.4-fold the genome of *H. gigas*, while our previous study, which was aimed at the mitogenomic analysis of this species (*H. gigas* humeral bone, which was collected in the mid-to-late 1800s, was used), did not allow the same[8].

The Steller's sea cow DNA reads had the main traits that were described for the historical DNA, particularly an increased frequency of apparent cytosine deamination substitutions compared to modern DNA) at the 5′-ends of the sequenced DNA fragments (Supplementary Figs. 1–4). At the same time, in the case of the petrous bone, we had an amazing preservation of DNA. All of the "historical" reads were used in the sequential analyses.

**Single-nucleotide polymorphisms (SNPs)-calling of Steller's sea cow specimen. Heterozygosity level.** The high-quality SNPs (with *p*-value < 0.05; coverage > 10×) were identified against the West Indian Manatee genome using the SnpEff software (v4.3)[17]. A total of 39,300,780 SNPs and 2,161,427 indels were found in total in the *H. gigas* genome (Dataset S1). The nonsense (high impact, i.e., completely disrupting the protein structure or leading to changes in its functions) as well as nonsynonymous mutations (moderate impact, i.e., variants that might change the protein effectiveness) were selected for the subsequent gene ontology (GO) analysis. In total, we described 8514 variants that had a high potential impact on the protein structure, and 30,956 variants that potentially changed the protein effectiveness (moderate impact). The genes name for the nonsense and nonsynonymous variants are presented in Dataset S2 and Dataset S3, respectively. GO analysis combined these genes into several gene categories related to the metabolic, immune, and hormone signaling pathways (Supplementary Table 3).

The average autosomal heterozygosity in *H. gigas* from the last population was 1.19 heterozygous sites per 1000 nucleotides (confidence interval (CI) 1.18–1.19). The heterozygosity of Steller's sea cow had an intermediate value between the values of the last and genetically inbred woolly mammoth population from Wrangel Island (the Middle Holocene) (1.00) and the juvenile Siberian woolly mammoth found in the Oimyakon District of Yakutia (the Late Pleistocene) (1.25)[18]. The closest sirenian species—dugong—has a higher heterozygosity level—2.19 (Fig. 2; Supplementary Table 4). Extant polar and brown bears have 0.4 and 1.7 heterozygous sites per 1000 nucleotides, respectively[19]. The same low heterozygosity levels were shown for other modern marine (Fig. 2) and terrestrial predators, such as wild cats and the Tasmanian devil[20,21]. It is presumed that predators have low population densities in comparison with herbivorous animals and as well as small effective population sizes[18].

**Positive selection analysis in Steller's sea cow genome.** The ratio of nonsynonymous substitutions to the rate of synonymous substitutions (dN/dS) is a good estimator of positive selection events that can happen in protein-coding genes. This value can be calculated as a number of nonsynonymous substitutions per nonsynonymous site to the number of synonymous substitutions per synonymous site. For homologous genes with a dN/dS ratio above 1, we can say that these genes evolved under positive selection (certain mutations might be advantageous). On the

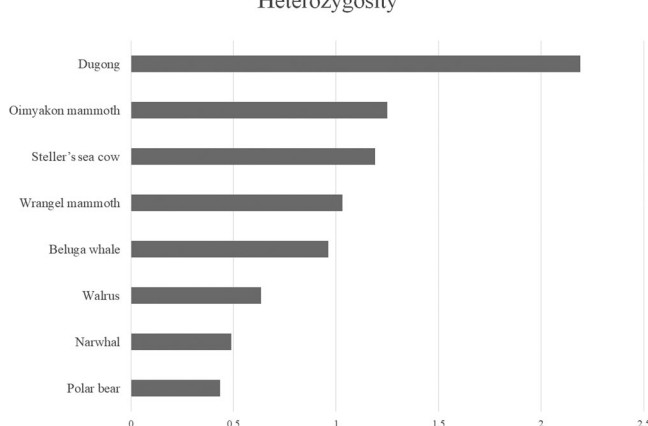

**Fig. 2 Average genome-wide autosomal heterozygosity values of Steller's sea cow and modern marine mammalian genomes.** The *X*-axis represents the average proportion of sites within the autosomes that are heterozygous. The *Y*-axis represents mammalian species.

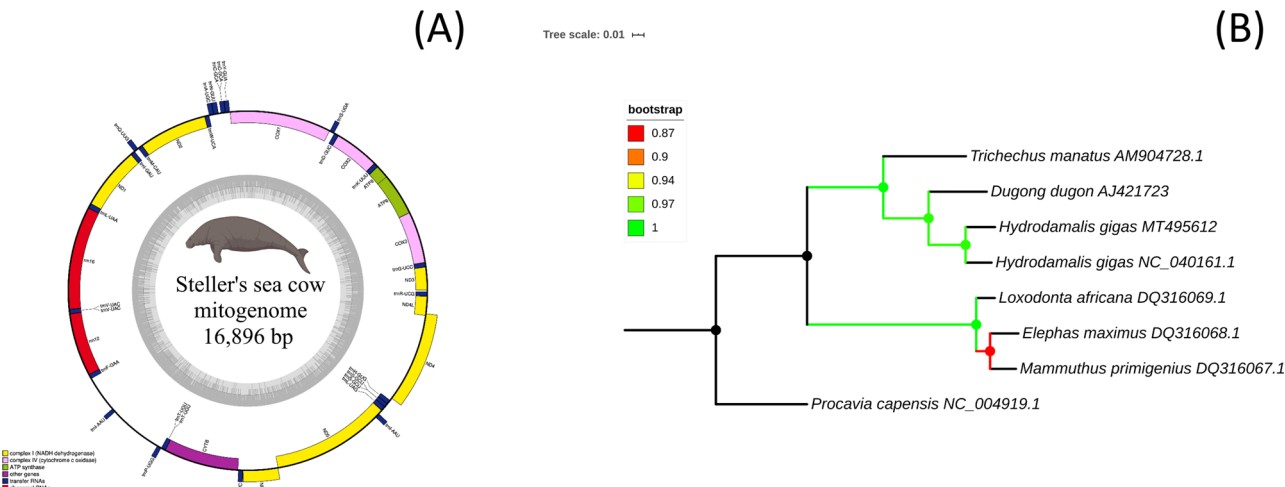

**Fig. 3 The mitochondrial genome of Steller's sea cow. A** Graphical map of the complete mitochondrial genome of Steller's sea cow with its gene features. Mitochondrial NADH dehydrogenase genes are marked by yellow color; mitochondrial cytochrome oxidase genes are marked by pink color; mitochondrial ribosomal RNA genes are marked by red color; mitochondrial ATP synthase genes are marked by green color; mitochondrial cytochrome B gene is marked by purple color; mitochondrial transfer RNA genes are marked by blue color. **B** Maximum likelihood phylogenetic tree reconstruction of Tethytheria species, including the extinct Steller's sea cow, based on their coding sequences. Bootstrap values are shown by color (from lower values—red to higher values—green).

other hand, for genes with a dN/dS ratio under 1, we can say that most of the mutations were neutral[22].

We obtained a list of the loci that were under positive selection in the *H. gigas* genome (Dataset S4). GO analysis combined these genes in several categories related to metabolic, immune, and hormone signaling pathways (e.g., GO:0046325—the negative regulation of glucose import; GO:0006952—a defense response; GO:0090278—the negative regulation of peptide hormone secretion; and GO:0048018—receptor ligand activity) (Supplementary Table 5). The genes that formed the negative regulation of the glucose import category (*PEA15*, *SIRT6*, *LEP*, and *VIMP*), were related to obesity and diabetes in human[23–26]. Surprisingly, the leptin (*LEP*) gene was under the positive selection in *H. gigas* as well as in other Pinnipedia and Cetacea species[13]. The tooth-related enamelin (*ENAM*) has been found as a gene that evolved under positive selection as also described previously[9]. GO analysis for genes with stricter dN/dS ratio (above three) categories were also found to be related to the defense response (GO:0006952) or GO:0048018—receptor ligand activity (Supplementary Table 6).

**De novo assembly of Steller's sea cow genome. The coverage of Steller's sea cow partial genome by *H. gigas* DNA reads**. To gain insight into the demographic structure, we performed de novo assembly for the *H. gigas* DNA reads. As a result, partial *H. gigas* genome assembly ("PRJNA484555"), which consists of 998,083 scaffolds was built with N50 equal to 1345 nucleotides, and the total assembly size was 1239 million bp. We used this assembly as a reference to describe depth of coverage for our *H. gigas* DNA reads. The total coverage of *H. gigas* partial genome was above 25.4-fold. This partial assembly was used then for the Pairwise Sequentially Markovian Coalescent (PSMC) analysis.

The mitochondrial DNA sequence of *H. gigas* was extracted from the genome assembly using blastn 2.7.1+. The mitogenome of *H. gigas* consists of 16,896 bp (GenBank accession number: MT495612) and includes 13 protein-coding genes (PCGs), 2 rRNA genes and 22 tRNA genes (Fig. 3A). In our previous study, we for the first time sequenced and analyzed the whole-mitochondrial genome of *H. gigas* museum specimen (14574)—humeral bone, which was collected in the mid-to-late 1800s by representatives of the Russian-American Company on Commander Islands[8].

The mitogenome for Steller's sea cow, which is published in this study, has the better quality but shows genetic similarity first one. Two *H. gigas* mitochondrial DNA sequences ("NC_040161.1" and MT495612 from this study) were used for phylogenetic reconstruction of the Tethytheria species based on their complete mitochondrial genomes.

**Sex determination of Steller's sea cow specimen**. The method described in Pečnerová et al.[27] and *H. gigas* sequencing dataset were used for sex determination of Ber/07/Yk-1 specimen. The coverage ratio of chrX/chr8 for the total data pool was 1.03, which means that Steller's sea cow specimen was a female. The number of *H. gigas* reads aligned on African elephant chromosomes and coverage percentage of each chromosome are presented in Supplementary Table 7.

**Phylogenetic analysis based on mitochondrial and nuclear genomes**. Phylogenetic reconstruction based on coding sequences (CDS) of mtDNA showed the "classical" topology for modern and extinct species from Tethytheria[8,9]. Proboscidea and Sirenia specimens have proper clustering according to their phylogenetic position and our *H. gigas* specimen as well as previously published[8], located in one cluster. The high bootstrap (>85%) support demonstrated the reliability of maximum likelihood clustering (Fig. 3B). Nuclear genome-based phylogeny also support this clustering (Supplementary Fig. 5).

**Demographic structure of Steller's sea cow**. The *H. gigas* partial genomic assembly obtained in this study was used to evaluate the demographic history of this animal in comparison with other extinct species and modern species that have successfully passed the period between Late Pleistocene to the Early Holocene. We used the Pairwise Sequentially Markovian Coalescent (PSMC) model, which is not suitable for past millennia but shows a demographic history of species or populations even based on one individual in the case of a high sequence coverage rate[28,29], for the Steller's sea cow demographic analysis (Fig. 1D) compared with dugong—*D. dugong*, two extinct Pleistocene species—two specimens of a woolly mammoth (*Mammuthus primigenius*) from Oimyakon (Yakutia, Russia) and

Wrangel Island (Chukotka, Russia)[18], and Pleistocene Lena horse (CGG10022)—*Equus lenensis* (Yakutia, Russia)[30].

The PSMC output for the extinct species showed that *H. gigas*, as well as its extant sirenian relative dugong, had only one bottleneck during its evolutionary history compared with the extinct terrestrial, herbivorous mammals. The woolly mammoth and Lena horse had at least two bottlenecks in the past hundreds of thousands of years (Fig. 1D).

To avoid the possible bias between terrestrial and marine mammals, we also conducted a comparative analysis of the demographic trajectory of Steller's sea cow and modern Arctic mammals, such as the beluga whale—*Delphinapterus leucas*, polar bear—*Ursus maritimus*, walrus—*Odobenus rosmarus*, and narwhal—*Monodon monoceros*[31].

We showed that the herbivorous Steller's sea cow had a different population history compared to modern marine predators: the beluga whale, polar bear, walrus, and narwhal. Possibly, the absence of drastic recent bottlenecks in the demographic trajectories of *D. leucas*, *U. maritimus*, *O. rosmarus*, and *M. monoceros* (Supplementary Fig. 7) relates to these species all being predators, which usually have smaller (compared with herbivorous) effective population sizes and low heterozygosity[18]. We presume that *H. gigas*, as well as other stenobiontic herbivorous marine mammals, were possibly more sensitive to climate change and the attendant sea-level rise, loss of feeding, and rookery sites.

## Discussion

Our study sheds light on the sirenian evolution and partially exonerates Paleolithic and modern humans, who found only a small fraction of once a large Steller's sea cow population with a relatively low level of heterozygosity. The average autosomal heterozygosity in *H. gigas* from the last population had an intermediate value between the values of the last and genetically inbred woolly mammoth population from Wrangel Island (the Middle Holocene) (1.00) and juvenile Siberian woolly mammoth found in the Oimyakon District of Yakutia (the Late Pleistocene) (1.25)[18]. This marine mammal was most likely a victim of significant climate changes during the Pleistocene and the associated environmental and biodiversity changes[32]. The sea level and temperature changes during the Late Pleistocene[33] may have negatively affected the *H. gigas* populations because they inhabited the coastal zone. A dramatic post-glacial sea-level rise near the Pleistocene–Holocene boundary[7] appears to have split the Steller's sea cow population.

Due to several morphological traits Steller's sea cows could not dive deep, so when the sea retreated and the temperature dropped, the number of suitable feeding sites for *H. gigas* decreased. *H. gigas* starvation and the associated fragmentation of species distribution possibly led to the origin of the small, separate refugia on the North Pacific islands for this mammal. Approximately 5000 years ago, the sea level stabilized; however, by this time, the range and Steller's sea cow population size had significantly decreased. Paleolithic hunter-gatherers also made their contribution to the *H. gigas* extinction, but, to be fair, only a few findings of Steller's sea cow have been described from dozens of Holocene coastal archeological sites (Supplementary Table 8) in the Bering Sea region[2,34–36].

The demographic history of *H. gigas* has a relatively different trajectory compared with modern Arctic marine mammals. However, we identified genes with apparent signatures of positive selection in Steller's sea cow, and most of them affect metabolic, immune, and hormone signaling pathways and are possibly related to anatomical and physiological adaptations of *H. gigas* to the Pleistocene–Holocene climatic perturbations. Some of these genes were also detected as loci under positive selection in extant marine mammal genomes. One of them, the leptin gene, which influences energy homeostasis and reproductive regulation, was under positive selection in Steller's sea cow as well as the Cetacea and Pinnipedia genomes[13]. We presume that this finding shows convergent evolution not only among modern marine species[37] but also between species that went extinct for several reasons.

Finally, we conclude that the population on the Commander Islands, which was the last surviving *H. gigas* population, like the last woolly mammoth population from Wrangel Island[18], was subject to reduced genetic diversity and possibly, despite the size (around 2000 animals[38]), was on the verge of extinction by the time of the arrival of the Vitus Bering's Great Northern Expedition.

## Methods

**Steller's sea cow specimen description**. The *H. gigas* petrous bone, field number: Ber/07/Yk-1 (Fig. 1C) was collected on Bering Island for the Museum of the World Ocean (Kaliningrad, Russia) exposition. The skull bones (including the petrous bone) preservation suggested that the animal died in the last years of the *H. gigas* population's existence on the Commander Islands (during the 1760s). It was impossible to conduct radiocarbon dating for this specimen due to the proximity of these dates to the present day[39].

In 2019, petrous bone among other bone material arrived at the Zoological Institute Russian Academy of Sciences (Saint-Petersburg, Russia) for the Steller's sea cow skeleton reconstruction. After identification, the petrous bone was placed in a labeled bag and sent to the National Research Center "Kurchatov Institute" (Moscow, Russia) for further historical DNA extraction and DNA-library preparation and sequencing.

**Historical DNA extraction and DNA-library preparation**. The petrous bone was treated with ultraviolet light for 20 min before the DNA extraction process. A dental drill was used for extremely careful sampling of bone powder from the petrous bone in the ancient DNA facilities of the National Research Center Kurchatov Institute (Moscow, Russia). After collection, the extracted sample was immediately used for historical DNA extraction in the same facility. Historical DNA from the *H. gigas* museum specimen was extracted from the bone powder following the silica beads methodology. This method is based on several sequential steps:

(1) Historical DNA isolation in buffer containing ethylenediaminetetraacetic acid (EDTA) and proteinase K;
(2) DNA enrichment on silica beads in the binding buffer; and
(3) DNA washing in ethanol and DNA elution.

Several independent DNA extractions were performed. As result, six multiplexed DNA-libraries were prepared using an Ovation® Ultralow Library System V2 kit (NuGEN, USA). The amplified DNA-libraries were quantified using a high-sensitivity chip on a 2100 Bioanalyser instrument (Agilent Technologies, USA). Multiple negative controls were used during the historical DNA extraction and DNA-library amplification. The negative controls did not contain DNA after the DNA extraction, and the DNA-libraries, which were prepared from the negative controls, were not amplified.

**Test DNA sequencing and PALEOMIX analysis**. The S2 flowcell of an Illumina Novaseq6000 genome analyzer (Illumina, USA) was used for the test (low-coverage) sequencing of *H. gigas* DNA-libraries with paired-end reads of 150 bp in length.

To remove contaminants from the sequencing data, we used the BBDuk software, which is included in the BBMap package v38.49 (www.sourceforge.net/projects/bbmap/), using the bacterial, fungal, plant, virus, and "other" databases (http://jgi.doe.gov/data-and-tools/bbtools/bb-tools-user-guide/). The output from the BBDuk tool v38.49 was processed through the PALEOMIX v1.2.14 pipeline[16], and mapping was done against the West Indian Manatee reference genome sequence ("TriManLat1.0 [https://www.ncbi.nlm.nih.gov/assembly/GCF_000243295.1/]") using BWA v0.7.17 under the "rescale" options[40].

The number of aligned reads representing the fraction of endogenous DNA varied from 26.07 to 47.11% for the test-sequencing DNA-libraries (Supplementary Table 1). Based on the endogenous DNA fraction in the sequenced DNA-libraries, four libraries (Lib1k; Lib3k, Lib4-2, and St1) were selected for the deep sequencing. The number of endogenous reads was calculated as the ratio between the number of mapped reads and the number of post-filtering reads (after PALEOMIX).

**Deep DNA sequencing and PALEOMIX analysis**. DNA-libraries with a higher content of endogenous DNA (Supplementary Table 2) were used for deep sequencing. For the deep DNA sequencing, we used an equimolar DNA-library pool. The S2 flowcell of the Illumina Novaseq6000 genome analyzer (Illumina,

USA) was used for the deep sequencing of the selected *H. gigas* DNA-libraries with paired-end reads of 50 bp in length.

To remove contaminants from the sequencing data, we used the BBDuk software, which is included in the BBMap package v38.49 (www.sourceforge.net/projects/bbmap/), using the bacterial, fungal, plant, virus, and "other" databases (http://jgi.doe.gov/data-and-tools/bbtools/bb-tools-user-guide/). The output from the BBDuk tool v38.49 was processed through the PALEOMIX 1.2.14 pipeline[16], and mapping was done against the West Indian Manatee reference genome sequence ("TriManLat1.0 [https://www.ncbi.nlm.nih.gov/assembly/GCF_000243295.1/]") using BWA v0.7.17 under the "rescale" options[40].

Postmortem DNA damage patterns were created using the MapDamage v2.0 tool[41]. We used MapDamage models to downscale the base quality scores according to the probability of being DNA damage by-products in order to reduce the impact of nucleotide misincorporations in the downstream analyses (Supplementary Figs. 1–4). The number of endogenous reads was calculated as the ratio between the number of mapped reads and the number of post-filtering reads (after PALEOMIX).

We also performed the same type of PALEOMIX analysis for Wrangel and Oimyakon woolly mammoths—*M. primigenius*[18], Lena horse (CGG10022 specimen)—*E. lenensis*[30], dugong—*D. dugon*, beluga whale—*D. leucas*, polar bear—*U. maritimus*, walrus—*O. rosmarus*, and narwhal—*M. monoceros*[31] (The National Center for Biotechnology Information (NCBI) accessions are presented in Supplementary Table 9). These additional genomic datasets were used in the comparative population size history analysis with Steller's sea cow using the Pairwise Sequentially Markovian Coalescent (PSMC) model (Fig. 1D; Supplementary Fig. 6; Supplementary Fig. 7).

**Single-nucleotide polymorphism (SNP) detection and genome-wide heterozygosity analysis.** The BAM files that were obtained using the PALEOMIX pipeline for Steller's sea cow, as well as for Wrangel and Oimyakon woolly mammoths, were used for SNP-calling with BCFtools (v1.9)[42] with a minimum base quality of 30 (–min-BQ parameter). Only SNPs with coverage higher than 10, and $p < 0.05$ were used in the subsequent analyses.

We used mlRho v2.9[43] to estimate the heterozygosity of modern mammals: dugong—*D. dugon*, beluga whale—*D. leucas*, polar bear—*U. maritimus*, walrus—*O. rosmarus*, and narwhal—*M. monoceros*; and extinct mammals: two specimens of woolly mammoth (*M. primigenius*) from Oimyakon (the Late Pleistocene) and Wrangel Island (the Middle Holocene)[18]; as well as the heterozygosity of Steller's sea cow—*H. gigas*. We calculated the population mutation rate (θ) estimate, which approximates the expected heterozygosity under the infinite sites model[18].

We filtered out bases using SAMtools v1.7[42] with quality below 30 and filtered out reads with a mapping quality below 30. We also excluded sites with a depth lower than 1/3 and higher than two times the estimated average coverage for each sample. The estimates from mlRho revealed the level of heterozygosity for *H. gigas* and the other animals mentioned above (Supplementary Table 4).

To annotate nonsynonymous nucleotide substitutions in coding regions for the *H. gigas* genomic data, we used SNPeff v4.3[17]. This analysis included several steps:

(1) Protein database development for the *T. manatus* ("TriManLat1.0 [https://www.ncbi.nlm.nih.gov/assembly/GCF_000243295.1/]") reference genome based on its annotation.
(2) Nonsynonymous (missense) variant identification—moderate or nondisruptive variants that might change the protein function.
(3) Nonsense (disruptive) variant identification—disruptive variants that might change the protein function (e.g., stop codons or splice donor variants).

To analyze the disruptive variants, we selected genes with nonsense mutations in their exon sequences. While the functional gene list analysis is only available for a restricted number of species, we converted the *H. gigas* gene IDs to the human gene names for functional analysis, using the GOrilla web-based application v1[44].

**Positive selection analysis in the Steller's sea cow genome.** Using the *H. gigas* VCF file as well as the *T. manatus* genome and its annotation, we performed an analysis to find genes that evolved under positive selection in the Steller's sea cow genome. SNPGenie v2019.10.31[45] software was used for this type of analysis.

Gene ontology analysis was conducted for *H. gigas* genes that evolved under positive selection, using the GOrilla web-based application v1[44].

**Steller's sea cow genome de novo assembly.** The sequencing output was also used for de novo assembly. To assemble the genome of *H. gigas*, we used all the sequencing datasets (Supplementary Table 2) after removing contaminants. The genome assembly was constructed using SPAdes (v3.10)[46]. For the SPAdes assembly, we added the -careful parameter, which minimizes the number of errors in the final contigs using the BayesHammer error corrector and automatic k-mer estimation[47].

The mitochondrial DNA sequence of *H. gigas* was extracted from the partial genome assembly using blastn v2.7.1+. The resulting consensus sequence was annotated using MITOS v2[48], and was visualized using the GeSeq web-interface v1.84[49].

**Sex determination of the Steller's sea cow specimen.** To determine the sex of the *H. gigas* specimen, we used the previously described method from Pečnerová et al.[27]. Briefly, we mapped the *H. gigas* endogenous reads to the African elephant reference genome ("Loxafr4 [ftp://ftp.broadinstitute.org/pub/assemblies/mammals/elephant/loxAfr4/]") with trimming of the PCR-duplicates, and then compared the depth of coverage between chromosomes—chr8 and chrX—which have comparable sizes. The numbers of reads was normalized out of the chromosome sizes. We expected to see an equal percentage of mapping to chr8 and chrX in the case of female and half less for chrX compared with chr8 in the case of male.

**Phylogenetic analyses of the extinct Steller's sea cow based on its complete mitochondrial genome.** The phylogenetic analysis of the mitochondrial DNA coding sequences (CDS) was performed for the Sirena species: Steller's sea cow—*H. gigas* ("NC_040161.1" and this study—MT495612), dugong—*D. dugon* ("AJ421723"), West Indian manatee—*T. manatus* ("AM904728") and for the Proboscides species: African savanna elephant—*Loxodonta africana* ("DQ316069"), Asian elephant—*Elephas maximus* ("DQ316068"), woolly mammoth—*M. primigenius* ("DQ316067"). The rock hyrax—*Procavia capensis* mitochondrial genome ("NC_004919.1") was added as an outgroup. We excluded the control region and transfer RNAs from the analysis because these mtDNA loci are not always suitable for phylogenetic reconstructions[50,51].

The maximum likelihood (ML) analysis was conducted using RAxML v8.2.12. Nodal support was evaluated using 1000 replications of rapid bootstrapping implemented in RAxML[52]. Phylogenetic tree reconstruction has been drawn in iTOL v4[53].

**Phylogeny of Tethytheria nuclear genome sequences, rooted with a rock hyrax outgroup.** We used BUSCO v4.0.5[54] to infer gene orthologues among Steller's sea cow—*H. gigas* ("PRJNA484555"), the West Indian manatee—*T. manatus* ("TriManLat1.0 [https://www.ncbi.nlm.nih.gov/assembly/GCF_000243295.1/]"), the African savanna elephant—*L. africana* ("Loxafr3.0 [https://www.ncbi.nlm.nih.gov/assembly/GCF_000001905.1/]"), the Wrangel woolly mammoth—*M. primigenius* (the mammoth genome was reconstructed using BCFtools v1.9[42] by creating a consensus sequence from the African savanna elephant genome ("Loxafr3.0 [https://www.ncbi.nlm.nih.gov/assembly/GCF_000001905.1/]"), and the rock hyrax—*P. capensis* ("ProCapCap_v2_BIUU_UCD [https://www.ncbi.nlm.nih.gov/assembly/GCA_004026925.2/]") genomes based on the Mammalia odb10 database (https://busco-data.ezlab.org/v4/data/). We found 908 complete mammalian orthologs common to all five genomes. Each of these groups of orthologs was first aligned one to one using MUSCLE v3.8.31 with a -maxiters parameter equal to 10[55]. The phylogenetic tree of the gene was reconstructed using the ML method with RAxML program[52]. We used a quick bootstrap analysis and searched for the most efficient ML tree with 1000 bootstraps. The best model with respect to the likelihood of a fixed tree was automatically determined using the GTRGAMMA option of RAxML. Then, we generated a species tree using the ASTRAL coalescent-based species tree estimation program v5.7.2[56].

**Mutation rate estimation.** The BAM files that were obtained using the PALEOMIX pipeline v1.2.14 for Steller's sea cow were used for the mutation rate estimation. We computed the pairwise distances between this species using a consensus base call in the ANGSD software suite v0.930 (parameters: -minQ 25 -minmapq 25 -uniqueonly 1 -remove_bads 1)[57]. The mutation rate per generation was calculated as:

$$\text{Mutation rate} = \text{pairwise distance} \times \text{generation time}/2 \times \text{divergence time}$$

Despite the fact that the exact time of divergence time between Trichechidae and Dugongidae is unclear and occurred a long time ago, we used 41.3 Mya as a split point based on Springer et al.[9]. We assumed a *H. gigas* generation time of 27 years. The generation time and mutation rate for the extinct *H. gigas*, *E. lenensis*, and two *M. primigenius* specimens as well as for the extant species, *D. dugon*, *D. leucas*, *U. maritimus*, *O. rosmarus*, and *M. monoceros* are presented in the Supplementary Table 10. The mutation rate index was used for the demography structure analysis of Steller's sea cow compared with extinct and modern mammal species (Supplementary Table 10).

**Demography structure analysis of Steller's sea cow.** We used the Pairwise Sequentially Markovian Coalescent (PSMC) model[28] to display the demographic history of Steller's sea cow (*H. gigas*) as well as two other extinct species, Pleistocene Lena horse (CGG10022 specimen)—*E. lenensis*[30] and two specimens of a woolly mammoth (*M. primigenius*) from Oimyakon and Wrangel Island[18], based on their diploid genome sequences.

We compared the demographic history of *H. gigas* with another sirenian species —dugong—*D. dugon* and modern Arctic marine species: beluga whale—*D. leucas*, polar bear—*U. maritimus*, walrus—*O. rosmarus*, and narwhal—*M. monoceros*. To analyze the support for the resultant PSMC analysis, we performed 100 bootstrap replicates (Supplementary Fig. S6) for *H. gigas* and modern mammals[31,58].

We used our genome assembly of *H. gigas* as well as reference genomes of extant marine mammals (Supplementary Table 9) for this type of analysis. The sex

chromosome scaffolds were removed from the assemblies for the sequential PSMC analysis. The method based on the PSMC model uses the distribution of heterozygous sites in the genome and a Markov model to reconstruct the history of the effective population size[28]. Consensus sequences were generated using the SAMtools v1.7 -mpileup command and the 'vcf2fq' command from vcfutils.pl[42].

The demography structure analysis was conducted in the PSMC software v0.6.5 (https://github.com/lh3/psmc) and was visualized using psmc_plot.pl. Filters for the base quality, mapping quality, root-mean-squared mapping quality below 30, and depth below 1/3 and higher than two times the average coverage estimated for each library were applied. We used the following parameters: number of iterations 25 and the atomic time interval "$4 + 25*2 + 4 + 6$.

**Reporting summary**. Further information on research design is available in the Nature Research Reporting Summary linked to this article.

## Data availability

The raw reads and mitochondrial and nuclear genome assemblies from Steller's sea cow are available for download through the National Center for Biotechnology Information, BioProject ID PRJNA484555. Accession numbers of previously published genomic data, which were reprocessed in this study, are available at Supplementary Material. All other data are included in the paper or available upon request. Source data are provided with this paper.

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

## Acknowledgements

We want to thank Anna Tikhonova for her valuable comments and English proofreading of this manuscript. We are also grateful to Ekaterina Khritonenkova for her artistic talent. We want to thank Prof. Jorge Galindo-Villegas for his valuable comments. This work was carried out using high-performance computing resources of the federal center for collective usage at the NRC Kurchatov Institute, http://computing.kiae.ru/. This work was supported by a Russian Foundation for Basic Research (RFBR), grants #18-00-00398 and #18-00-00399. This study was completed within the framework of the Federal themes of the Zoological Institute no. AAAA-A19-119032590102-7 "Phylogeny, morphology, and systematics of placental mammals". This work was partially carried out in the Kurchatov Center for Genome Research and supported by the Ministry of Science and Higher Education of Russian Federation, grant #075-15-2019-1659. The funders had no role in the study design, data collection and analysis, decision to publish, or preparation of the manuscript. Nord University Open Access Fund covers the OA publication costs.

## Author contributions

Conceptualization: F.S.S., S.M.R., A.N.T., and A.V.N.; data curation: F.S.S; formal analysis: F.S.S, A.V.N., and S.M.R.; funding acquisition: A.N.T. and A.V.N.; investigation: E.S.B., S.V.T., and N.V.S; methodology: E.S.B., S.V.T., and N.V.S; project administration: A.N.T. and A.V.N.; resources: F.S.S. and S.M.R.; software: F.S.S. and S.M.R.; supervision: A.N.T. and A.V.N.; visualization: F.S.S., D.A.A., and A.A.K.; writing, original draft: A.V.N.; writing, review, and editing: all authors.

## Competing interests

The authors declare no competing interests.
