## [Peer Review File · Nature Communications]

Reviewers' Comments:

Reviewer #1:

Remarks to the Author:

Review of "Stellar sea cow, what made it go extinct?" By Sharko et al, For Nat. Comms.

This paper describes the authors' extraction and sequencing of a whole ancient genome from the recently extinct Stellar sea cow, *H. gigas*, with 10.8x coverage. They carry out a PSMC analysis, together with previously sequenced ancient genomes, (mammoth and Pleistocene horse) to show that, unlike these non marine mammals, the major loss of diversity occurred before 400kya, long before humans are hypothesised to have reached populations of *H. gigas*. The authors thus conclude that the last population of *H. gigas* was subject to reduced genetic diversity and possibly on the edge of extinction by the time sailors reached it in the late 1700s.

Minor comment

There are examples of slightly odd language use and colloquialisms throughout which could do with editing - e.g.

Abstract Line 28 "launched"

Abstract Line 31 "began to extinct"

Main text Line 53 "latest" should be "last"

Main text Line 61 "stroke" should be "strike"

Main text Line 64 "figure out"

And so on, plus grammar, arbitrary use of "the" etc.

Major comment

The Stellar sea cow is a fascinating creature, and its nuclear genome has not yet been sequenced. While sequencing one individual high quality ancient genome can be considered a reasonable amount for a paper, and the authors seem to do a thorough job in terms of lab work, this paper is let down by the fact that they only seem to perform/report one real analysis of interest (estimating heterozygosity, and carrying out a PSMC analysis, although a couple of (previously published?) trees are also included in supplementary). It seems so much more could be gleaned from the genome after all their work generating it.

For example... functional genes? Given that there is a whole genome for the related manatee, could they examine any of the genes of interest between the two taxa, as a recent paper did for the ENAM gene in relation to the difference in teeth/mouth parts and feeding across Sirenia. E.g. perhaps genes involved in cold adaptation? Size? Other adaptations between the different sea cow environments? Are there any genes of interest under selection? It would also be interesting to know the sex of the individual.

For the PSMC analysis, where they examine genetic diversity through time in comparison to extinct land mammals, it would perhaps be interesting to compare the levels in genetic diversity with other related marine mammals, such as the manatee? Or, alternatively, other heavily predated northern hemisphere marine mammals? e.g. Walrus? Whales? Seals? The drop in diversity before 400 kya is hypothesised to arise from changes in climate, these taxa might provide more information regarding this.

Finally, the authors hypothesise that the population of *H. gigas* might have been on its last (metaphorical) legs, this is something that could perhaps be investigated further? Unless more

genomes of Stellar Seacow can be obtained, inbreeding calculations from ROH (runs of homozygosity) are probably not possible, but perhaps they might be able to examine potential measures of genetic load across the genome, particularly in comparison to manatee? For example, van der Valk et al (2019) Current Biology paper on Gorilla describes several methods to estimate genetic load, some of which should be possible without closely related genomes available, e.g. the number of mutations in genomic regions that are conserved across several vertebrates (i.e. any mutations are thus likely to be deleterious). I don't know how feasible this would be with the available data, but it would certainly be worth looking into.

All these analyses or any further analyses describing the genome content would make the paper much more worthy of publication in Nature Communications. As it stands, I would recommend that this paper is rejected with encouragement to resubmit if further analyses are undertaken to increase the scope of the paper.

Reviewer #2:

Remarks to the Author:

This paper describes an attempt to recover the nuclear genome of Steller's sea cow (*H. gigas*). The researchers recover a substantial portion of nuclear genome (~1.58 Gb), and show high congruence in the evolutionary relationships among Afrotheria using both the mitochondrial and nuclear DNA. The authors also use the nuclear data in a PSMC analysis to examine the change in *H. gigas* effective population size over time and note a strong decline that begins in the mid to early Pleistocene. Nuclear genomes of extinct animals are still in short supply due to the difficulty in recovering such large targets from degraded remains. In my experience these projects always generate interest in the palaeogenetic community both due to the findings in the original paper and also in the wealth of data that is made available for subsequent projects. I have little doubt that data will prove invaluable in future studies looking at functional adaptations between Sirenia species, particularly as all remaining species tend to be found in much warmer conditions.

I have little doubt in the validity of the data, and don't think the paper over-extrapolates its results. The fact that the authors observe strong correspondence in both their nuclear and mitochondrial phylogenies and everything falls where expected, is a strong indication that they are truly recovering what they describe. I suspect they may recover more information or obtain better resolution in their PSMC analysis if a larger portion of the genome was recovered, but I have little doubt in the results of the PSMC as a whole, as both the Wrangel and Oimyakon mammoths included in the analysis recover very similar trajectories to previously published PSMC analyses.

I don't think there are any major flaws in the analysis that prohibit publication though I feel like there are segments where I'm questioning why exactly the authors chose to work with the data a certain way or what exactly was used in the analyses at certain steps. None of the analyses seem necessarily wrong, but I feel like there could be more explanation of the rationale behind some of the items below.

1. There appear to be some differences between the initial "test Illumina" run and the "deep Illumina" run based on the data in Tables S1 and S2, and described in the paragraph beginning on line 148. In the paper I was primarily surprised by the increase in endogenous DNA to 55% when the pooled libraries maxed out at 47%. If pooled evenly as the paper suggests, I would expect the final pool to have a concentration somewhere in the same range as the constituent libraries, while the final pool is a fair bit higher. Although possible to have been caused by variation, pipetting errors, or other minor missteps, it still seemed odd.

However, looking at the SOM it appears that the overall composition of the libraries between the two runs is different. In addition to the aforementioned endogenous increases being greater than I would expect, it appears the average read length of reads drastically decreased for all four chosen libraries from an average of 139.63 bp to 54.47 bp. I'm not sure what would cause such a drastic shift within previously sequenced libraries short of manually size-selecting a different fragment range or using a different read length, but no such information is mentioned in the paper or the SOM.

2. It was also unclear to me specifically whether the aligned or de novo data was used for the phylogenies and PSMC data. I assume it was the de novo data (at least for the PSMC), but feel this should be made more clear in the paper.

3. On line 179, there is mention of defining partitions here in a section of the mitochondrial genome. Based on its position in the paper I assume this is only for the mitochondrial DNA and the nuclear data is not partitioned? If this is not the case then that needs to be clarified. Additionally, I feel like this statement might fit better when describing the mitochondrial phylogenies in the section beginning on line 201.

4. Additionally for the mitochondrial phylogenies (line 201), I think the authors should clarify specifically what they used for the analysis and how it was prepared. As written I assume, the authors removed all 13 identified CDS's and appended them end to end as their input. While I think this is a valid approach, I feel this should be mentioned explicitly as it deviates from the simplest solution of just analyzing the entire mitochondrial genome.

5. On line 171 it mentions that ~783M paired-end reads were used for genome assembly, and I'm not entirely sure where this number comes from. The paragraph before mentions a decontamination program, but it's also unclear to me what exactly is going in as input (whether just the "deep Illumina" libraries or combined with the "test Illumina" ones). A simple table in the SOM here with each input library, and the number of reads going in and coming out of the decontamination steps would be a good thing to have in the SOM.

6. I noticed that the authors BioProject was already released on NCBI, and took a look at the summary statistics. I'm not sure why (and if the data on NCBI hasn't been updated yet), but the data presented there does not match what's described in the paper. For example on NCBI, the total sequence length is reported as 1,239,941,981 bp vs 1,577,000,000 in the paper. Likewise, the number of scaffolds is 998,083 in NCBI's global statistics table and 1,018,345 reported in the paper. The scaffold N50 and genome coverage are also incorrect between the two: 1435/10.8x in the paper, and 1430/11x on NCBI.

7. My last point is relatively minor, but I feel should be corrected for transparency's sake. The depth of coverage of the genome is reported as 10.8x, but this is the value of the mapped data against *T. manatus*. If the de novo data is to be published on NCBI a better way of showing the depth, might be to map all reads back to the de novo contigs and report the average depth there.

For the most part the methodology is sound. The authors include enough information that I'm reasonably confident of the integrity of the data, and addressing the issues/questions above I feel will make people even more confident. The authors might want to explicitly mention (either in the main text or the SOM) that the amazing preservation of their specimen is also supported by their large fragment lengths (especially if the "test" Illumina ones remain consistent) and the low level of damage at the termini of their reads, as seen in their MapDamage plots.

The only thing I think the authors should mention is some description of their blanks. Extraction blanks to monitor for the introduction of human (or other exogenous DNA) are still a must in the field, and I feel especially in projects where new genomes from previously unsequenced species are produced. This could also be coupled with an expanded section on the decontamination program as requested above, to examine what may have been introduced following excavation.

In general there's enough information that I think most people within the ancient DNA field could be able to reproduce the results. I think a little additional data on the extraction of DNA might be useful as in-solution silica based extraction protocols seem to have somewhat fallen out of favour in comparison to column-based methods. Additionally, citation 24 (line 129) is cited for explaining the extraction of ancient samples, but the cited paper Orlando et al. (2013) doesn't actually seem to provide details in the main text, and instead cites an earlier paper by the same group – Orlando et al. (2011).

With respect to the rest I see no major issues. I am not too familiar with the PALEOMIX pipeline, but from a quick look at the documentation it seems like a reasonable choice.

Lastly, I wanted to point out that the paper is written fairly well, but the wording in a few sections jumped out at me as strange. I've listed these below for consideration. The paper will need to be split into the sections wanted by Nat Comms though this should be relatively simple to do as the paper already has a very natural break for this at starting at line 63.

Line 22: "... and present days human race is a witness..." is an odd phrase. I would consider just cutting that entirely and reworking the sentence slightly.

Line 59: The paragraph beginning here seems at odds both with itself, with what is mentioned in the abstract (line 26), and earlier in the Introduction (Line 46). I suspect the authors are trying to say that climate and Palaeolithic hunting may have reduced population levels, prior to terminal extinction by sailors, but this paragraph should be reworked slightly to make this clearer.

Line 80: Peak seems like a weird word to use here. Maybe bottomed out would be better, or something else to indicate this is when the population crashed stopped/slowed down.

Emil Karpinski

H_Gigas_1.0

Organism name: Hydrodamalis gigas (Steller's sea cow)

Isolate: HGIGA-B-2019

BioSample: SAMN15314634

BioProject: PRJNA484555

Submitter: National Research Center Kurchatov Institute

Date: 2020/07/08

Assembly level: Scaffold

Genome representation: full

RefSeq category: representative genome

GenBank assembly accession: GCA_013391785.1 (latest)

RefSeq assembly accession: n/a

RefSeq assembly and GenBank assembly identical: n/a

WGS Project: JACANZ01

Assembly method: SPAdes v. 3.10

Expected final version: no

Genome coverage: 11.0x

Sequencing technology: Illumina NovaSeq

IDs: 7332621 [UID] 20284928 [GenBank]

History (Show revision history)

Global statistics

Total sequence length	1,239,941,981
Total ungapped length	1,238,988,409
Gaps between scaffolds	0
Number of scaffolds	998,083
Scaffold N50	1,430
Scaffold L50	265,168
Number of contigs	1,091,214
Contig N50	1,345
Contig L50	281,348
Total number of chromosomes and plasmids	0
Number of component sequences (WGS or clone)	998,083

Reviewer #3:

Remarks to the Author:

In this study, Sharko and colleagues sequence and analyse the genome from a 250-year-old specimen of Steller's sea cow, an extinct sirenian mammal. They reconstruct the demographic history of the species from the diploid genome using the pairwise sequential Markovian coalescent (PSMC) method. Based on the results of this analysis, the authors conclude that the species experienced a substantial and prolonged decline in population size well before they were encountered and hunted by modern humans.

The sequencing and assembly of the nuclear genome of Steller's sea cow is an impressive achievement, and the genome will undoubtedly be a useful resource for researchers. However, I have some concerns about the study and am not entirely convinced that the conclusions are supported by the analyses. I have a range of suggestions for improving the manuscript, including some additional analyses as well as further comparisons that will make the study more comprehensive.

Abstract

(1) Please add a brief description of the data analyses, such as PSMC and the comparisons of heterozygosity.

Main

(2) line 41: For additional context here, maybe describe the past diversity of Sirenia and whether any other sirenian species or lineages have gone extinct in the Pleistocene and Holocene.

(3) line 49: It would be helpful to show the estimated geographic range of Steller's sea cow in Figure 1B, including Japan and the Pacific coast of North America.

(4) line 53: When did the species become restricted to the Commander Islands? It would be useful to have this information here.

(5) line 59: This paragraph is important for interpreting the results of the demographic analyses and should be expanded to provide more detail.

(6) line 65: Please give more information about the specimen that was used for genome sequencing, such as its exact age and how it was dated.

(7) line 70: What was different about the specimen used for the previous mitogenomic analysis? Was it more poorly preserved, was it older, or was it a different skeletal element?

(8) line 72: The genome coverage of 10.8x is not high enough for accurate calling of heterozygous sites for PSMC analysis (Nadachowska-Brzyska et al., 2016). Does using different quality thresholds affect the demographic reconstruction? What is the proportion of missing data? What is the size of the genome, and what proportion is included in the genome sequence? These details need to be reported so that readers can have a better idea of the reliability of the PSMC results.

(9) line 74: How were these two species chosen for comparison? It would be interesting to compare the demographic reconstructions for other species, whether they are based on whole genomes (e.g., PSMC) or mitochondrial DNA (e.g., skyline plots).

(10) line 80: The interpretation depends on the accuracy of the timescale on the PSMC plot. How was the plot scaled to time – do you have accurate estimates of generation time and mutation rate? A bootstrapping analysis should also be conducted to estimate the uncertainty in the reconstruction.

(11) line 90: The confidence interval for the heterozygosity seems very short. How is this influenced by different quality filtering thresholds? The average coverage across the genome is quite low, which would have a negative influence on estimating heterozygosity.

(12) line 91: Comparisons with heterozygosity in a wider range of species would be highly informative.

(13) line 101: This section is quite speculative and it would instead be helpful to include some paleoclimatic reconstructions of the habitat in northern Pacific Ocean and Beringia.

(14) line 111: PSMC plots are unable to provide resolution for very recent timeframes so it is not possible to comment on the population size of Steller's sea cow over the past few thousand years. The shape of the PSMC plot can also be affected by changes in population structure (Mazet et al. 2015) – are you able to exclude this factor?

Methods

(15) line 164: How were the PSMC plots for the woolly mammoths and Lena horse rescaled? Do you have accurate estimates of the generation times and mutation rates?

(16) line 177: What was the mitochondrial DNA sequence used for? There is no description of this in the Main part of the manuscript.

(17) line 198: Please provide justification for the PSMC settings.

(18) line 201: This section describes phylogenetic analyses of the mitochondrial genome and nuclear genome sequences, but these analyses and their results are not mentioned at all in the Main part of the manuscript. They do not seem very relevant to the study and the tree in Figure S6 does not seem to be useful (it simply shows the two sirenians grouping together and the two proboscideans grouping together). If you choose to retain these analyses, please report the results more prominently and incorporate them into the Main part of the manuscript.

Figures

(19) Please add a scale bar or gridlines (latitude/longitude) to the map in panel B.

References

Mazet O., Rodríguez W., Chikhi L. (2015) Demographic inference using genetic data from a single individual: Separating population size variation from population structure. *Theoretical Population Biology*, 104: 46-58.

Nadachowska-Brzyska K., Burri R., Smeds L., Ellegren H. (2016) PSMC analysis of effective population sizes in molecular ecology and its application to black-and-white *Ficedula* flycatchers. *Molecular Ecology*, 25: 1058-1072.

Reviewer #4:

Remarks to the Author:

This is a very straightforward topic and I applaud the authors for not trying to get more out of the data than can be supported based on the sample size. I have one major comment however - it seems strange to me that the comparative analyses of heterozygosity and PSMC are limited to 2 land mammals, when there is plenty of marine mammal data out there, that I think would be more relevant. I mean, is there any reason to expect that land mammal patterns are replicated in marine mammal patterns? I strongly suggest they consider this paper

<https://www.sciencedirect.com/science/article/pii/S2589004219300896>

On a narwhale genome, in which the authors actually do these analyses on a range of relevant species.

So in summary, while I have no qualms about the data or analysis on the sea cow genome, the comparative analyses should be expanded.

Apart from that the English needs a gentle polishing, many small grammatical errors.

RESPONSE TO THE EDITOR AND REVIEWERS

We would like to thank you for your consideration, valuable and friendly comments, and suggestions to improve this manuscript. As suggested by the review panel, considerable changes on the abstract, introduction, results, and discussion sections have been performed for easy reading and clear understanding of the manuscript. Besides, the English language and scientific writing were improved by a professional English interpreter as well as a native speaker.

REVIEWER COMMENTS

Reviewer #1 (Remarks to the Author):

Review of "Stellar sea cow, what made it go extinct?" By Sharko et al, For Nat. Comms.

This paper describes the authors' extraction and sequencing of a whole ancient genome from the recently extinct Stellar sea cow, *H. gigas*, with 10.8x coverage. They carry out a PSMC analysis, together with previously sequenced ancient genomes, (mammoth and Pleistocene horse) to show that, unlike these non marine mammals, the major loss of diversity occurred before 400kya, long before humans are hypothesised to have reached populations of *H. gigas*. The authors thus conclude that the last population of *H. gigas* was subject to reduced genetic diversity and possibly on the edge of extinction by the time sailors reached it in the late 1700s.

Minor comment

There are examples of slightly odd language use and colloquialisms throughout which could do with editing - e.g.

Abstract Line 28 "launched"

Abstract Line 31 "began to extinct"

Main text Line 53 "latest" should be "last"

Main text Line 61 "stroke" should be "strike"

Main text Line 64 "figure out"

And so on, plus grammar, arbitrary use of "the" etc.

Major comment

The Stellar sea cow is a fascinating creature, and its nuclear genome has not yet been sequenced. While sequencing one individual high quality ancient genome can be considered a reasonable amount for a paper, and the authors seem to do a thorough job in terms of lab work, this paper is let down by the fact that they only seem to perform/report one real analysis of interest (estimating heterozygosity, and carrying out a PSMC analysis, although a couple of (previously published?) trees are also included in supplementary). It seems so much more could be gleaned from the genome after all their work generating it.

For example... functional genes? Given that there is a whole genome for the related manatee, could they examine any of the genes of interest between the two taxa, as a recent paper did for the ENAM gene in relation to the difference in teeth/mouth parts and feeding across Sirenia. E.g. perhaps genes involved in cold adaptation? Size? Other adaptations between the different sea cow environments? Are there any genes of interest under selection? It would also be interesting to know the sex of the individual.

For the PSMC analysis, where they examine genetic diversity through time in comparison to extinct land mammals, it would perhaps be interesting to compare the levels in genetic diversity with other related marine mammals, such as the manatee? Or, alternatively, other heavily predated northern hemisphere marine mammals? e.g. Walrus? Whales? Seals? The drop in diversity before 400 kya is hypothesised to arise from changes in climate, these taxa might provide more information regarding this.

Finally, the authors hypothesise that the population of *H. gigas* might have been on its last (metaphorical) legs, this is something that could perhaps be investigated further? Unless more genomes of Stellar Seacow can be obtained, inbreeding calculations from ROH (runs of homozygosity) are probably not possible, but perhaps they might be able to examine potential measures of genetic load across the genome, particularly in comparison to manatee? For example, van der Valk et al (2019) Current Biology paper on Gorilla describes several methods to estimate genetic load, some of which should be possible without closely related genomes available, e.g. the number of mutations in genomic regions that are conserved across several vertebrates (i.e. any mutations are thus likely to be deleterious). I don't know how feasible this would be with the available data, but it would certainly be worth looking into.

All these analyses or any further analyses describing the genome content would make the paper much more worthy of publication in Nature Communications. As it stands, I would recommend that this paper is rejected with encouragement to resubmit if further analyses are undertaken to increase the scope of the paper.

Answers to Reviewer #1:

Thank you for your suggestions. We completely agree that the manuscript should be clear and free from grammar mistakes. A native English speaker from the USA as well as a professional English interpreter reviewed our manuscript. All of the your corrections were approved and added.

The main object of this study is clarifying the reasons for Steller's sea cow extinction, but we totally agree that the additional analyses related to describing loci that were under positive selection are important. We conducted additional Dn/Ds analysis and Gene ontology analysis for *H. gigas* genomic data (L137 – L152), as well as we discriminated the sex of this animal (L173 – L178). We also analyzed the demographic history of *H. gigas* in comparing not only with extinct terrestrial mammals but also with modern marine mammals such as narwhal, beluga, walrus, polar bear and etc (L187 – L211) (modern marine mammals from this paper: PMID: 31054839). We suppose that further studies based not only on specimen from the Commander Islands will clarify the detailed history of this species.

We found the list of the genes which were under positive selection in *H. gigas* genome (L137 – L152). Among them we found *ENAM* gene, as well as a list of the genes related to metabolic, immune, and hormone signaling pathways; one of them *leptin* has previously been described as evolutionary important for Pinnipedia and Cetacea species (PMID: 22046310). Based on these data, we also speculate that there is a convergent evolution not only among modern marine species but also between extinct marine species (such as Steller's sea cow).

Based on your suggestion, we also added information about heterozygosity in modern mammals and included suitable references for detailed information (L128 – 136).

We suppose that for genetic load analysis we need more Steller's sea cow genomes with good coverage. We tried to conduct the same type of analysis as Valk et al (2019) in Current Biology paper but only two specimens (Steller's sea cow and manatee) analyzed as well as the differences in genome coverage did not allow us to make such a comparison to manatee. We hope, that in our (or our colleagues) further studies this part will be clarified.

Reviewer #2 (Remarks to the Author):

This paper describes an attempt to recover the nuclear genome of Steller's sea cow (*H. gigas*). The researchers recover a substantial portion of nuclear genome (~1.58 Gb), and show high congruence in the evolutionary relationships among Afrotheria using both the mitochondrial and nuclear DNA. The authors also use the nuclear data in a PSMC analysis to examine the change in *H. gigas* effective population size over time and note a strong decline that begins in the mid to early Pleistocene.

Nuclear genomes of extinct animals are still in short supply due to the difficulty in recovering such large targets from degraded remains. In my experience these projects always generate interest in the palaeogenetic community both due to the findings in the original paper and also in the wealth of data that is made available for subsequent projects. I have little doubt that data will prove invaluable in future studies looking at functional adaptations between Sirenia species, particularly as all remaining species tend to be found in much warmer conditions.

I have little doubt in the validity of the data, and don't think the paper over-extrapolates its results. The fact that the authors observe strong correspondence in both their nuclear and mitochondrial phylogenies and everything falls where expected, is a strong indication that they are truly recovering what they describe. I suspect they may recover more information or obtain better resolution in their PSMC analysis if a larger portion of the genome was recovered, but I have little doubt in the results of the PSMC as a whole, as both the Wrangel and Oimyakon mammoths included in the analysis recover very similar trajectories to previously published PSMC analyses.

I don't think there are any major flaws in the analysis that prohibit publication though I feel like there are segments where I'm questioning why exactly the authors chose to work with the data a certain way or what exactly was used in the analyses at certain steps. None of the analyses seem necessarily wrong, but I feel like there could be more explanation of the rationale behind some of the items below.

1. There appear to be some differences between the initial "test Illumina" run and the "deep Illumina" run based on the data in Tables S1 and S2, and described in the paragraph beginning on line 148. In the paper I was primarily surprised by the increase in endogenous DNA to 55% when the pooled libraries maxed out at 47%. If pooled evenly as the paper suggests, I would expect the final pool to have a concentration somewhere in the same range as the constituent libraries, while the final pool is a fair bit higher. Although possible to have been caused by variation, pipetting errors, or other minor missteps, it still seemed odd.

However, looking at the SOM it appears that the overall composition of the libraries between the two runs is different. In addition to the aforementioned endogenous increases being greater than I would expect, it appears the average read length of reads drastically decreased for all four chosen libraries from an average of 139.63 bp to 54.47 bp. I'm not sure what would cause such a drastic shift within previously sequenced libraries short of manually size-selecting a different fragment range or using a different read length, but no such information is mentioned in the paper or the SOM.

2. It was also unclear to me specifically whether the aligned or *de novo* data was used for the phylogenies and PSMC data. I assume it was the *de novo* data (at least for the PSMC), but feel this should be made more clear in the paper.

3. On line 179, there is mention of defining partitions here in a section of the mitochondrial genome. Based on its position in the paper I assume this is only for the mitochondrial DNA and the nuclear data is not partitioned? If this is not the case then that needs to be clarified. Additionally, I feel like this statement might fit better when describing the mitochondrial phylogenies in the section beginning on line 201.

4. Additionally for the mitochondrial phylogenies (line 201), I think the authors should clarify specifically what they used for the analysis and how it was prepared. As written I assume, the authors removed all 13 identified CDS's and appended them end to end as their input. While I think this is a valid approach, I feel this should be mentioned explicitly as it deviates from the simplest solution of just analyzing the entire mitochondrial genome.

5. On line 171 it mentions that ~783M paired-end reads were used for genome assembly, and I'm not entirely sure where this number comes from. The paragraph before mentions a decontamination program, but it's also unclear to me what exactly is going in as input (whether just the "deep Illumina" libraries or combined with the "test Illumina" ones). A simple table in the SOM here with each input library, and the number of reads going in and coming out of the decontamination steps would be a good thing to have in the SOM.

6. I noticed that the authors BioProject was already released on NCBI, and took a look at the summary statistics. I'm not sure why (and if the data on NCBI hasn't been updated yet), but the data presented there does not match what's described in the paper. For example on NCBI, the total sequence length is reported as 1,239,941,981 bp vs 1,577,000,000 in the paper. Likewise, the number of scaffolds is 998,083 in NCBI's global statistics table and 1,018,345 reported in the paper. The scaffold N50 and genome coverage are also incorrect between the two: 1435/10.8x in the paper, and 1430/11x on NCBI.

7. My last point is relatively minor, but I feel should be corrected for transparency's sake. The depth of coverage of the genome is reported as 10.8x, but this is the value of the mapped data against *T. manatus*. If the *de novo* data is to be published on NCBI a better way of showing the depth, might be to map all reads back to the *de novo* contigs and report the average depth there.

For the most part the methodology is sound. The authors include enough information that I'm reasonably confident of the integrity of the data, and addressing the issues/questions above I feel will make people even more confident. The authors might want to explicitly mention (either in the main text or the SOM) that the amazing preservation of their specimen is also supported by their large fragment lengths (especially if the "test" Illumina ones remain consistent) and the low level of damage at the termini of their reads, as seen in their MapDamage plots.

The only thing I think the authors should mention is some description of their blanks. Extraction blanks to monitor for the introduction of human (or other exogenous DNA)

are still a must in the field, and I feel especially in projects where new genomes from previously unsequenced species are produced. This could also be coupled with an expanded section on the decontamination program as requested above, to examine what may have been introduced following excavation.

In general there's enough information that I think most people within the ancient DNA field could be able to reproduce the results. I think a little additional data on the extraction of DNA might be useful as in-solution silica based extraction protocols seem to have somewhat fallen out of favour in comparison to column-based methods. Additionally, citation 24 (line 129) is cited for explaining the extraction of ancient samples, but the cited paper Orlando et al. (2013) doesn't actually seem to provide details in the main text, and instead cites an earlier paper by the same group – Orlando et al. (2011).

With respect to the rest I see no major issues. I am not too familiar with the PALEOMIX pipeline, but from a quick look at the documentation it seems like a reasonable choice.

Lastly, I wanted to point out that the paper is written fairly well, but the wording in a few sections jumped out at me as strange. I've listed these below for consideration. The paper will need to be split into the sections wanted by Nat Comms though this should be relatively simple to do as the paper already has a very natural break for this at starting at line 63.

Line 22: "... and present days human race is a witness..." is an odd phrase. I would consider just cutting that entirely and reworking the sentence slightly.

Line 59: The paragraph beginning here seems at odds both with itself, with what is mentioned in the abstract (line 26), and earlier in the Introduction (Line 46). I suspect the authors are trying to say that climate and Palaeolithic hunting may have reduced population levels, prior to terminal extinction by sailors, but this paragraph should be reworked slightly to make this clearer.

Line 80: Peak seems like a weird word to use here. Maybe bottomed out would be better, or something else to indicate this is when the population crashed stopped/slowed down.

Emil Karpinski

Answers to Reviewer #2

1. Thank you for your comments. We used different Illumina sequencing kits for the test- and deep sequencing (2×150 bp and 2 ×50 bp, respectively) for a number of nonscientific reasons. Additional explanations were added to the M&Ms section (L276 – L309) and Supplementary file (Table S1-S2). The differences between endogenous DNA containing in two sequencing runs can also be related to a relatively higher percentage of mapping for shorter reads.
2. We used *de novo* assembled *H. gigas* genome for PSMC analysis. The main reason to make *H. gigas* genome *de novo* assembly was demography analysis. Additional explanations were added to the M&Ms section (L409 – L413).
3. We significantly improved manuscript text that describes phylogenetic analysis based on mitochondrial DNA and nuclear genome. We hope that this part looks better now (L362 – L386).
4. More detailed information about mitochondrial DNA assembly, annotation, and phylogenetic analysis is shown in M&Ms: “Steller’s sea cow genome *de novo* assembly” (L349 – L351) and “Phylogenetic analyses of the extinct Steller's sea cow based on it complete mitochondrial genome” (L362 – L372) sections. Briefly, we used 13 mitochondrial protein-coding genes for phylogenetic analysis based on mitogenomes. Nuclear genomes clustering based on mammalian gene orthologs common to all 5 genomes (mammoth, manatee, Steller’s sea cow, hyrax, elephant) was made separately.
5. The number of contaminant reads for the test- and deep sequencing runs were included in Supplementary file (please see, Table S1 and Table S2).
6. Thank you for your attention to details. The mapping coverage to the *T. manatus* genome was 10.8X, but we decided to use (as you recommended below) mapping coverage (25.4 X) to *H. gigas* reference genome *de novo* assembled in this study. Also, we checked all of the output values in the manuscript again.
7. Thank you for your suggestion. We mapped our DNA reads against *de novo* assembled genome of *H. gigas*. As result, we have 25.4X coverage of the genome. We suggest that 25.4X coverage is more reliable and suitable for PSMC analysis.

We added information about negative controls, that were used during historical DNA extraction and DNA-libraries amplification (M&Ms section) (L260 – L274). We also included more details about DNA extraction (L273 – L274) and changed citation from Orlando et al. (2013) to Orlando et al. (2011). The manuscript was split into the sections recommended by Nature Communications. We added additional types of analyses which you and other reviewers kindly recommended. Also, we check English grammar again and added your corrections.

Reviewer #3 (Remarks to the Author):

In this study, Sharko and colleagues sequence and analyse the genome from a 250-year-old specimen of Steller's sea cow, an extinct sirenian mammal. They reconstruct the demographic history of the species from the diploid genome using the pairwise sequential Markovian coalescent (PSMC) method. Based on the results of this analysis, the authors conclude that the species experienced a substantial and prolonged decline in population size well before they were encountered and hunted by modern humans.

The sequencing and assembly of the nuclear genome of Steller's sea cow is an impressive achievement, and the genome will undoubtedly be a useful resource for researchers. However, I have some concerns about the study and am not entirely convinced that the conclusions are supported by the analyses. I have a range of suggestions for improving the manuscript, including some additional analyses as well as further comparisons that will make the study more comprehensive.

Abstract

(1) Please add a brief description of the data analyses, such as PSMC and the comparisons of heterozygosity.

Main

(2) line 41: For additional context here, maybe describe the past diversity of Sirenia and whether any other sirenian species or lineages have gone extinct in the Pleistocene and Holocene.

(3) line 49: It would be helpful to show the estimated geographic range of Steller's sea cow in Figure 1B, including Japan and the Pacific coast of North America.

(4) line 53: When did the species become restricted to the Commander Islands? It would be useful to have this information here.

(5) line 59: This paragraph is important for interpreting the results of the demographic analyses and should be expanded to provide more detail.

(6) line 65: Please give more information about the specimen that was used for genome sequencing, such as its exact age and how it was dated.

(7) line 70: What was different about the specimen used for the previous mitogenomic analysis? Was it more poorly preserved, was it older, or was it a different skeletal element?

(8) line 72: The genome coverage of 10.8x is not high enough for accurate calling of heterozygous sites for PSMC analysis (Nadachowska-Brzyska et al., 2016). Does using different quality thresholds affect the demographic reconstruction? What is the

proportion of missing data? What is the size of the genome, and what proportion is included in the genome sequence? These details need to be reported so that readers can have a better idea of the reliability of the PSMC results.

(9) line 74: How were these two species chosen for comparison? It would be interesting to compare the demographic reconstructions for other species, whether they are based on whole genomes (e.g., PSMC) or mitochondrial DNA (e.g., skyline plots).

(10) line 80: The interpretation depends on the accuracy of the timescale on the PSMC plot. How was the plot scaled to time – do you have accurate estimates of generation time and mutation rate? A bootstrapping analysis should also be conducted to estimate the uncertainty in the reconstruction.

(11) line 90: The confidence interval for the heterozygosity seems very short. How is this influenced by different quality filtering thresholds? The average coverage across the genome is quite low, which would have a negative influence on estimating heterozygosity.

(12) line 91: Comparisons with heterozygosity in a wider range of species would be highly informative.

(13) line 101: This section is quite speculative and it would instead be helpful to include some paleoclimatic reconstructions of the habitat in northern Pacific Ocean and Beringia.

(14) line 111: PSMC plots are unable to provide resolution for very recent timeframes so it is not possible to comment on the population size of Steller's sea cow over the past few thousand years. The shape of the PSMC plot can also be affected by changes in population structure (Mazet et al. 2015) – are you able to exclude this factor?

Methods

(15) line 164: How were the PSMC plots for the woolly mammoths and Lena horse rescaled? Do you have accurate estimates of the generation times and mutation rates?

(16) line 177: What was the mitochondrial DNA sequence used for? There is no description of this in the Main part of the manuscript.

(17) line 198: Please provide justification for the PSMC settings.

(18) line 201: This section describes phylogenetic analyses of the mitochondrial genome and nuclear genome sequences, but these analyses and their results are not mentioned at all in the Main part of the manuscript. They do not seem very relevant to the study and the tree in Figure S6 does not seem to be useful (it simply shows the two sirenians grouping together and the two proboscideans grouping together). If you choose to retain these analyses, please report the results more prominently and incorporate them into the Main part of the manuscript.

Figures

(19) Please add a scale bar or gridlines (latitude/longitude) to the map in panel B.

References

Mazet O., Rodríguez W., Chikhi L. (2015) Demographic inference using genetic data from a single individual: Separating population size variation from population structure. *Theoretical Population Biology*, 104: 46-58.

Nadachowska-Brzyska K., Burri R., Smeds L., Ellegren H. (2016) PSMC analysis of effective population sizes in molecular ecology and its application to black-and-white *Ficedula* flycatchers. *Molecular Ecology*, 25: 1058-1072.

Answers to Reviewer #3:

1. Thank you for your consideration and valuable comments. The abstract was modified (L23 – L36).
2. The section devoted to sirenian distribution during the Pleistocene – Holocene period was added to Introduction (L42 – L53).
3. Figure 1B was changed.
4. Sirenian distribution area (as well as *H. gigas*) significantly changed and fragmented at the transition from the Late Pleistocene to the Early Holocene. This information was added to the Introduction section (L51 – L53).
5. We inserted additional information about climatic changes and sea-level rise at the transition from the Late Pleistocene to the Early Holocene in the Introduction section. Paper related to human activity (sea otter hunter) during the XVIII century was also discussed in the Introduction part (L51 – L53 and L68 – L72).
6. Information about Steller’s cow specimen was expanded. The skull bones (including petrous bone) preservation suggested that the animal died in the last years of sea cow population existence on Commander Islands (during the 1760s). In the case of this specimen, it was impossible to conduct radiocarbon dating for this specimen due to the proximity of these dates to the present day (L247 – L258). Using genome sequencing we showed that this animal was male (L173 – L178).
7. In the first paper (PMID: 30352279) we used humeral bone from the Zoological Institute of the Russian Academy of Sciences (St. Petersburg, Russian Federation) – museum specimen (14574). The first data were suitable for mtDNA reconstruction, but the percentage of *H. gigas* nuclear DNA was very low (less than 1%), because that we decide to use “fresh” petrous bone from another specimen which was found on Commander Islands recently. Here, we showed again that this type of skull bone is a treasury for historical/ancient DNA analysis. Description of previous specimen was added to Results (L166 – L169) section.
8. Thank you for your suggestion. The mapping coverage to the *T. manatus* genome was 10.8X, but we decided to use mapping coverage (25.4 X) to *H. gigas* reference genome *de novo* assembled in this study. We suggest that 25.4X coverage is more reliable and suitable for PSMC analysis. We used such filtering for PSMC: base quality, mapping quality and root-mean-squared mapping quality below 30, and depth below 1/3 and higher than 2-times the average coverage estimated for each library were applied (L414 – L418). Also previous studies showed that the PSMC can be conducted even based on one chromosome (PMID: 21753753), in case of our study the size of assembly equals to 1,239 million bp (L156 – L159).
9. Based on you and other reviewer suggestions, we conducted comparative PSMC analysis not only for extinct woolly mammoths and Lena horse but also for the modern marine mammals such as narwhal, beluga, walrus, polar bear and etc (L187 – L211). The demographic history does not change in any case for *H. gigas*. Interestingly, that extinct species were herbivorous, while modern marine species are predators. In our manuscript, we also discuss this as well as differences in heterozygosity level between herbivores and predators (L127 – L136).
10. Thank you for your suggestion. The estimation of mutation rates subsection was added to Material and Methods (L387 – L398). Mutation rates are presented in Table S9. Generation times are also presented in Table S9 (this information was received from

PMID: 25913407 and PMID: 31054839. To analyze the support for the resultant PSMC analysis we performed 100 bootstrap replicates for *H. gigas* and modern Arctic animals (Figure S6A-S6E) as well as was previously conducted for woolly mammoth and Lena horse (25913407 and PMID: 25512547, respectively)

11. These data are consistent with the mammoth manuscript. The mapping coverage to the *T. manatus* genome was 10.8X, but we decided (based on reviewer's suggestions) to use mapping coverage (25.4 X) to *H. gigas* reference genome *de novo* assembled in this study. We suggest that 25.4X coverage is more reliable and suitable for PSMC analysis.
12. We added information about heterozygosity extinct and modern mammals. Suitable references with detailed information and comparisons were also included (L127 – L136).
13. Paper related to human activity (sea otter hunter) during the XVIII century was also discussed in the Introduction part (L51 – L53 and L68 – L72). Changes in the sirenian distribution area at the transition from the Late Pleistocene to the Early Holocene were also described in the Introduction section (L42 – L50).
14. You are definitely right! PSMC does not work over the past few thousand years. But the main conclusion of our manuscript is *H. gigas* population had only one catastrophic population decline and it was bottomed out around 400 thousand years ago, and Steller's sea cow population has not been restored since then. Based on Steller's publication we know that the population size of the last Steller's sea cow population was around 2000 animals.
15. We added data for woolly mammoths and Lena horse, referring to the calculations published previously (PMID: 25913407 and PMID: 25512547, respectively)
16. Mitochondrial genome of *H. gigas* was used for phylogenetic analysis of Tethytheria species. The result of this analysis described in Results: "Phylogenetic analysis based on mitochondrial and nuclear genomes" section (L189 – L185).
17. We used the same settings as well as in narwhale genome paper (PMID: 31054839) (L408-L09).
18. The main object of this study is clarifying the reasons for Steller's sea cow extinction, but we decided to add phylogenetic analyses for our mitochondrial and nuclear data to show the confidence of historical DNA extraction, DNA sequencing, *de novo* assembly analysis.
19. The gridlines were added to Figure 1B.

Also we cited important paper by Mazet et al., 2015 as well as Nadachowska-Brzyska et al., 2016.

Reviewer #4 (Remarks to the Author):

This is a very straightforward topic and I applaud the authors for not trying to get more out of the data than can be supported based on the sample size. I have one major comment however - it seems strange to me that the comparative analyses of heterozygosity and PSMC are limited to 2 land mammals, when there is plenty of marine mammal data out there, that I think would be more relevant. I mean, is there any reason to expect that land mammal patterns are replicated in marine mammal patterns? I strongly suggest they consider this paper

<https://www.sciencedirect.com/science/article/pii/S2589004219300896>

On a narwhale genome, in which the authors actually do these analyses on a range of relevant species.

So in summary, while I have no qualms about the data or analysis on the sea cow genome, the comparative analyses should be expanded.

Apart from that the English needs a gentle polishing, many small grammatical errors.

Answers to Reviewer #4:

Thank you for your suggestions. We analyzed demographic history of *H. gigas* in comparing not only with extinct terrestrial mammals but also with modern marine mammals such as narwhal, beluga, walrus, polar bear and etc. (from this paper: PMID: 31054839) as you recommended (L187 – L211).

Based on your and other reviewers' comments we significantly improved the manuscript structure. The main improvement is adding positive selection analysis (L136 – L153) and Gene Ontology analysis for *H. gigas* genomic data (L116 – L126). Also, we discriminated the sex of this animal (male) (L173 – L178).

English language and scientific writing were improved by a professional interpreter and native speaker.

Reviewers' Comments:

Reviewer #1:

Remarks to the Author:

I feel the authors have done a good job in addressing all my queries. In combination with the other reviewer recommendations, I feel this has much improved the manuscript. The wider range of analyses of the *H.gigas* genome provides a more detailed picture of this species' biology, evolutionary history and extinction.

The only outstanding analysis, which I would have been interested to see, is a comparison of the heterozygosity through time between *H.gigas* and its closest relative - the manatee (and perhaps the inclusion of the marine mammal heterozygosity combined plot in the main text).

Reviewer #2:

Remarks to the Author:

I am happy to see that most of my changes have been addressed in this new revision. It was also exciting to read the new analyses the authors undertook with respect to identifying genes under selection. However, there are still a few minor and one or two larger issues I feel should be addressed before publication. I have commented on each of the authors' changes below as well as identifying any additional points of uncertainty. The most important among these would be some more details on the de novo assembly (#5), results of the negative controls (#8), and some questions I have about the sexing analysis. As I mention towards the end, there are still also quite a few spelling and grammar mistakes throughout, although I suspect most of these will be caught during subsequent steps. Overall, I'm happy to see how the paper is coming along, and this is should be ready with one more round of revisions.

Previous comments (Authors' responses in red and numbered; my additional points in black and preceded by "Response:"):

1. Thank you for your comments. We used different Illumina sequencing kits for the test- and deep sequencing (2×150 bp and 2 ×50 bp, respectively) for a number of nonscientific reasons. Additional explanations were added to the M&Ms section (L276 – L309) and Supplementary file (Table S1-S2). The differences between endogenous DNA containing in two sequencing runs can also be related to a relatively higher percentage of mapping for shorter reads.

Response: The data makes much more sense in light of the extra details. One small minor addition I would like to see here is a few words on how endogenous % was calculated, even if only in the caption of Tables S1 and S2. Based on the numbers given it appears to be calculated as [# mapped reads]/[# reads post-filtering], which is perfectly fine (and I think the right choice), but its sometimes in the field also represented as [# mapped reads]/[total sequenced read pairs], or, when programs like BLAST are used, as the [total number of taxa-of-interested identified reads]/[total classified reads]. Assuming I'm correct on the math above, I would literally just add something to the effect of: "Endogenous % calculated as the number of mapped reads (following map quality, size, and deduplication filtering) over the total number of reads retained after PALEOMIX quality filtration."

2. We used de novo assembled *H. gigas* genome for PSMC analysis. The main reason to make *H. gigas* genome de novo assembly was demography analysis. Additional explanations were added to the M&Ms section (L409 – L413).

Response: I'm okay with the changes done here.

3. We significantly improved manuscript text that describes phylogenetic analysis based on

mitochondrial DNA and nuclear genome. We hope that this part looks better now (L362 – L386).

Response: I'm alright with these changes, and it certainly makes the analysis clearer. I did notice that any mentions of partitioning the data is now gone from the manuscript, so I assume this was redone from scratch without partitioning?

4. More detailed information about mitochondrial DNA assembly, annotation, and phylogenetic analysis is shown in M&Ms: "Steller's sea cow genome de novo assembly" (L349 – L351) and "Phylogenetic analyses of the extinct Steller's sea cow based on its complete mitochondrial genome" (L362 – L372) sections. Briefly, we used 13 mitochondrial protein-coding genes for phylogenetic analysis based on mitogenomes. Nuclear genomes clustering based on mammalian gene orthologs common to all 5 genomes (mammoth, manatee, Steller's sea cow, hyrax, elephant) was made separately.

Response: I'm okay with the changes done here.

5. The number of contaminant reads for the test- and deep sequencing runs were included in Supplementary file (please see, Table S1 and Table S2).

Response: It's still unclear to me what exactly is being used for the de novo assembly. It appears this number has changed from the original draft (~783 M) to ~887 M in the current draft, but I'm still unclear as to which reads these specifically are. The closest I can get is ~884 M by taking the number of mapped reads from the deep sequencing run (adding the ones from the test-Illumina run brings this value too high). I think instead of listing the number of reads on L345, it would be clearer to say which specific reads were used and how they were treated. To clarify, if the mapped reads of all four libraries from the deep sequencing run were used, I would write something to that effect.

Apologies if I missed the contamination column in the table originally.

6. Thank you for your attention to details. The mapping coverage to the *T. manatus* genome was 10.8X, but we decided to use (as you recommended below) mapping coverage (25.4 X) to *H. gigas* reference genome de novo assembled in this study. Also, we checked all of the output values in the manuscript again.

Response: These seem to be all correct now.

7. Thank you for your suggestion. We mapped our DNA reads against de novo assembled genome of *H. gigas*. As result, we have 25.4X coverage of the genome. We suggest that 25.4X coverage is more reliable and suitable for PSMC analysis.

Response: I agree. I would recommend just one more small change to make it abundantly clear what this value is referring to (as two different methods of genome reconstruction are used in the paper): in both the abstract (L29-30) and the first time the 25.4 statistic is used in the paper (L73) the genome is referred to as a de novo genome.

8. We added information about negative controls, that were used during historical DNA extraction and DNA-libraries amplification (M&Ms section) (L260 – L274).

Response: I would like to see a little more information here on the results of the blanks themselves. Were they sequenced or not? If not, why not? And if they were sequenced what is the amount of reads that makes it through all the various mapping stages and the identity of any de novo contigs generated? A lot of this information could be added to directly to Tables S1 and S2 for comparison with the specimen libraries.

9. We also included more details about DNA extraction (L273 – L274) and changed citation from Orlando et al. (2013) to Orlando et al. (2011). The manuscript was split into the sections recommended by Nature Communications. We added additional types of analyses which you and other reviewers kindly recommended. Also, we check English grammar again and added your corrections.

Response: Overall, I'm alright with the changes here. There's still a few spelling and grammar mistakes throughout the paper, but I suspect these will all be caught over the next round of revisions/final copy-editing.

New Comments:

The only new item I find myself slightly confused by is the sex-determination. The authors use a bovine chromosome X sequence and a human chromosome Y sequence, and look at the difference in de novo contigs that aligned against these two sequences. Looking at Table S6, I'm unclear what exactly the coverage value presented here relates to (i.e. % coverage of the chromosomes or depth of coverage), and I'm curious why the authors didn't just map the data instead. Additionally I'm confused by the choice of X and Y reference sequences. Even if good assemblies with clearly identified chromosomes are not available for anything with Sirenia (a taxa I'm not as familiar with), there is a fairly good nuclear genome available for *Loxodonta africana* (LoxAfr v4; available for download here: <ftp://ftp.broadinstitute.org/pub/assemblies/mammals/elephant/loxAfr4/>). This genome was generated from a female (so there is no Y-chromosome sequence), but a method for approximating from a female-source reference has been previously described (see Pečnerová et al (2017) below) using the ratio of reads mapped to the X chromosome and autosome 8. I think this would be a much better analysis and uses a more closely related taxa (albeit still not ideal).

Pečnerová, P., et al. (2017). Genome-Based Sexing Provides Clues about Behavior and Social Structure in the Woolly Mammoth. *Current Biology*, 27 (22), P3505-P3510.

Reviewer #3:

Remarks to the Author:

Sharko and colleagues have extensively revised their study of a genome from a 250-year-old specimen of Steller's sea cow. The revised paper includes an analysis of selection and a strengthened PSMC analysis. Most of my previous comments have been addressed but I still have some concerns and suggestions.

Abstract

(1) Please add a concluding sentence that mentions the general implications of the results.

Main

(2) line 39: the first paragraph of the Introduction does not fit very well.

(3) line 51: this paragraph partly repeats a few points mentioned in the previous paragraph.

(4) line 144: for detecting genes under positive selection, a more conservative threshold should be used. Of the 5708 genes under 'positive selection' in Dataset S4, more than half have a dN/dS between 1 and 1.5. A more conservative threshold would be dN/dS >3 or >5.

(5) line 191: Mazet et al. 2015 is cited here but the paper does not discuss the possible impacts of population structure on the PSMC plot. This possibility needs to be mentioned.

(6) line 392: the divergence time between Trichechidae and Dugongidae is very deep when considering the short timescale of the demographic history of Steller's sea cow. The authors should include a strong caution about the uncertainty in the mutation rate estimate, which is important for the scale of the PSMC plot.

(7) line 406: the bootstrap results should preferably be added to Figure 1D.

Reviewer #4:

Remarks to the Author:

Thank you for your edits, which meet the concerns I raised.

RESPONSE TO THE EDITOR AND REVIEWERS

We would like to thank you for your consideration, valuable and friendly comments, and suggestions to improve this manuscript. Based on reviewers suggestions we created an additional figure which shows the difference in heterozygosity between extinct and extant mammals. We also estimated sex of the Steller's sea cow specimen based on a powerful method from Pečnerová et al., 2017. **Corrections are marked by yellow in manuscript and supplementary file.** Queries are in black, our rebuttal in blue.

REVIEWER COMMENTS

Reviewer #1 (Remarks to the Author):

I feel the authors have done a good job in addressing all my queries. In combination with the other reviewer recommendations, I feel this has much improved the manuscript. The wider range of analyses of the *H.gigas* genome provides a more detailed picture of this species' biology, evolutionary history and extinction.

The only outstanding analysis, which I would have been interested to see, is a comparison of the heterozygosity through time between *H.gigas* and its closest relative - the manatee (and perhaps the inclusion of the marine mammal heterozygosity combined plot in the main text).

It was impossible to use manatee genomic data for PSMC analysis because we used the manatee genome as a reference. We decided to use the freshly published *Dugong dugon* dataset (DRR251525) for such analysis. Thus, we added modern marine mammals (including dugong) and Steller's sea cow into PSMC plot (please refer to Supplementary Figure S7). As suspected, the dugong has the same demographical history as *H. gigas*. This is possibly related to global temperatures and sea-level rise during Late Pleistocene – Early Holocene transition (please refer to the Introduction section). We also provided heterozygosity data for dugong, Steller's sea cow, woolly mammoth, and other modern marine predators. Dugong showed the highest heterozygosity level between herbivorous animals (please refer to Figure 2).

Reviewer #2 (Remarks to the Author):

I am happy to see that most of my changes have been addressed in this new revision. It was also exciting to read the new analyses the authors undertook with respect to identifying genes under selection. However, there are still a few minor and one or two larger issues I feel should be addressed before publication. I have commented on each of the authors' changes below as well as identifying any additional points of uncertainty.

The most important among these would be some more details on the de novo assembly (#5), results of the negative controls (#8), and some questions I have about the sexing analysis. As I mention towards the end, there are still also quite a few spelling and grammar mistakes throughout, although I suspect most of these will be caught during subsequent steps. Overall, I'm happy to see how the paper is coming along, and this should be ready with one more round of revisions.

Thank you again for your important suggestions.

Previous comments (Authors' responses in red and numbered; my additional points in black and preceded by "Response:"):

1. Thank you for your comments. We used different Illumina sequencing kits for the test- and deep sequencing (2×150 bp and 2 ×50 bp, respectively) for a number of nonscientific reasons. Additional explanations were added to the M&Ms section (L276 – L309) and Supplementary file (Table S1-S2). The differences between endogenous DNA containing in two sequencing runs can also be related to a relatively higher percentage of mapping for shorter reads.

Response: The data makes much more sense in light of the extra details. One small minor addition I would like to see here is a few words on how endogenous % was calculated, even if only in the caption of Tables S1 and S2. Based on the numbers given it appears to be calculated as [# mapped reads]/[# reads post-filtering], which is perfectly fine (and I think the right choice), but its sometimes in the field also represented as [# mapped reads]/[total sequenced read pairs], or, when programs like BLAST are used, as the [total number of taxa-of-interested identified reads]/[total classified reads]. Assuming I'm correct on the math above, I would literally just add something to the effect of: "Endogenous % calculated as the number of mapped reads (following map quality, size, and deduplication filtering) over the total number of reads retained after PALEOMIX quality filtration."

This information was added to the Material and Methods section. Number of endogenous reads was calculated as a ratio between the total number of reads and the number of post-filtering reads (after PALEOMIX).

2. We used de novo assembled *H. gigas* genome for PSMC analysis. The main reason to make *H. gigas* genome de novo assembly was demography analysis. Additional explanations were added to the M&Ms section (L409 – L413).

Response: I'm okay with the changes done here.

Thank you.

3. We significantly improved manuscript text that describes phylogenetic analysis based on mitochondrial DNA and nuclear genome. We hope that this part looks better now (L362 – L386).

Response: I'm alright with these changes, and it certainly makes the analysis clearer. I did notice that any mentions of partitioning the data is now gone from the manuscript, so I assume this was redone from scratch without partitioning?

We analyzed phylogeny of Tethytheria based on mitochondrial and nuclear genomes separately. But for nuclear genome phylogenetic analysis we used all data. Possibly we did not describe it clearly for the first time.

4. More detailed information about mitochondrial DNA assembly, annotation, and phylogenetic analysis is shown in M&Ms: “Steller’s sea cow genome de novo assembly” (L349 – L351) and “Phylogenetic analyses of the extinct Steller’s sea cow based on its complete mitochondrial genome” (L362 – L372) sections. Briefly, we used 13 mitochondrial protein-coding genes for phylogenetic analysis based on mitogenomes. Nuclear genomes clustering based on mammalian gene orthologs common to all 5 genomes (mammoth, manatee, Steller’s sea cow, hyrax, elephant) was made separately.

Response: I'm okay with the changes done here.

Thank you.

5. The number of contaminant reads for the test- and deep sequencing runs were included in Supplementary file (please see, Table S1 and Table S2).

Response: It's still unclear to me what exactly is being used for the de novo assembly. It appears this number has changed from the original draft (~783 M) to ~887 M in the current draft, but I'm still unclear as to which reads these specifically are. The closest I can get is ~884 M by taking the number of mapped reads from the deep sequencing run (adding the ones from the test-Illumina run brings this value too high). I think instead of listing the number of reads on L345, it would be clearer to say which specific reads were used and how they were treated. To clarify, if the mapped reads of all four libraries from the deep sequencing run were used, I would write something to that effect.

Apologies if I missed the contamination column in the table originally.

Thank you for the comment. We used all the sequencing datasets (Table S2) after removing contaminants.

6. Thank you for your attention to details. The mapping coverage to the T. manatus genome was 10.8X, but we decided to use (as you recommended below) mapping coverage (25.4 X) to H. gigas reference genome de novo assembled in this study. Also, we checked all of the output values in the manuscript again.

Response: These seem to be all correct now.

Thank you.

7. Thank you for your suggestion. We mapped our DNA reads against de novo assembled genome of *H. gigas*. As result, we have 25.4X coverage of the genome. We suggest that 25.4X coverage is more reliable and suitable for PSMC analysis.

Response: I agree. I would recommend just one more small change to make it abundantly clear what this value is referring to (as two different methods of genome reconstruction are used in the paper): in both the abstract (L29-30) and the first time the 25.4 statistic is used in the paper (L73) the genome is referred to as a de novo genome.

Thank you for your suggestion. These corrections were made in the Abstract and Introduction.

8. We added information about negative controls, that were used during historical DNA extraction and DNA-libraries amplification (M&Ms section) (L260 – L274).

Response: I would like to see a little more information here on the results of the blanks themselves. Were they sequenced or not? If not, why not? And if they were sequenced what is the amount of reads that makes it through all the various mapping stages and the identity of any de novo contigs generated? A lot of this information could be added to directly to Tables S1 and S2 for comparison with the specimen libraries.

Thank you for your suggestion. We actively include the negative controls during our experiments. Sometimes, they help us to discard poor samples/sample sets (primarily during ancient human studies). In the case of Steller's sea cow DNA extraction, we also used negative controls, but they did not contain DNA after DNA extraction, and DNA-libraries from the negative controls were not amplified.

9. We also included more details about DNA extraction (L273 – L274) and changed citation from Orlando et al. (2013) to Orlando et al. (2011). The manuscript was split into the sections recommended by Nature Communications. We added additional types of analyses which you and other reviewers kindly recommended. Also, we check English grammar again and added your corrections.

Response: Overall, I'm alright with the changes here. There's still a few spelling and grammar mistakes throughout the paper, but I suspect these will all be caught over the next round of revisions/final copy-editing.

Thank you. Now the manuscript was edited in English proof-reading service.

New Comments:

The only new item I find myself slightly confused by is the sex-determination. The authors use a bovine chromosome X sequence and a human chromosome Y sequence, and look at the difference in de novo contigs that aligned against these two sequences. Looking at Table S6, I'm unclear what exactly the coverage value presented here relates

to (i.e. % coverage of the chromosomes or depth of coverage), and I'm curious why the authors didn't just map the data instead. Additionally I'm confused by the choice of X and Y reference sequences. Even if good assemblies with clearly identified chromosomes are not available for anything with Sirenia (a taxa I'm not as familiar with), there is a fairly good nuclear genome available for *Loxodonta africana* (LoxAfr v4; available for download here: <ftp://ftp.broadinstitute.org/pub/assemblies/mammals/elephant/loxAfr4/>). This genome was generated from a female (so there is no Y-chromosome sequence), but a method for approximating from a female-source reference has been previously described (see Pečnerová et al (2017) below) using the ratio of reads mapped to the X chromosome and autosome 8. I think this would be a much better analysis and uses a more closely related taxa (albeit still not ideal).

Pečnerová, P., et al. (2017). Genome-Based Sexing Provides Clues about Behavior and Social Structure in the Woolly Mammoth. *Current Biology*, 27 (22), P3505-P3510.

First, thank you for this very important suggestion. Previously, we used a method described in Westbury et al., 2019 , to save the calculation logic for further analyses. Based on your suggestion we decided to use a much more powerful method from Pečnerová et al., 2017. We tested the coverage ratio of chrX/Chr 8 for *H. gigas* genomic data, and the coverage ratio of chrX/Chr 8 was 1.03, from which we can conclude that the specimen is female.

Reviewer #3 (Remarks to the Author):

Sharko and colleagues have extensively revised their study of a genome from a 250-year-old specimen of Steller's sea cow. The revised paper includes an analysis of selection and a strengthened PSMC analysis. Most of my previous comments have been addressed but I still have some concerns and suggestions.

Abstract

- (1) Please add a concluding sentence that mentions the general implications of the results.**

Concluding sentence was added.

Main

- (2) line 39: the first paragraph of the Introduction does not fit very well.**

The first paragraph of the Introduction was deleted from the article.

- (3) line 51: this paragraph partly repeats a few points mentioned in the previous paragraph.**

These two paragraphs were combined and re-written.

(4) line 144: for detecting genes under positive selection, a more conservative threshold should be used. Of the 5708 genes under ‘positive selection’ in Dataset S4, more than half have a dN/dS between 1 and 1.5. A more conservative threshold would be dN/dS >3 or >5.

Usually dN/dS threshold >1 is considered as a benchmark (PMID: 19081788). Nevertheless, we got 685 genes with dN/dS threshold >3. Gene ontology analysis found several categories for these genes related to a defense response and signaling pathways. We also added it to the Supplementary Material.

(5) line 191: Mazet et al. 2015 is cited here but the paper does not discuss the possible impacts of population structure on the PSMC plot. This possibility needs to be mentioned.

Thank you for the comment. We added the main point of Mazet et al (2015) paper, that even one individual can be used for PSMC analysis. In our M&Ms section, we describe the parameters of PSMC. We used single individual parameters from Westbury et al (2019).

(6) line 392: the divergence time between Trichechidae and Dugongidae is very deep when considering the short timescale of the demographic history of Steller’s sea cow. The authors should include a strong caution about the uncertainty in the mutation rate estimate, which is important for the scale of the PSMC plot.

Even though the exact time of divergence time between Trichechidae and Dugongidae is unclear and happened a long time ago, we used 41.3 Mya as a split point based on Springer et al., 2015 (PMID: 26050523).

(7) line 406: the bootstrap results should preferably be added to Figure 1D.

Figure 1D was changed. We added bootstrap as well as dugong demography. Thank you.

Reviewer #4 (Remarks to the Author):

Thank you for your edits, which meet the concerns I raised.

Thank you for your contribution!

Reviewers' Comments:

Reviewer #1:

Remarks to the Author:

I am happy with the authors responses to my final query.

Reviewer #2:

Remarks to the Author:

I think the authors have done a great job incorporating the changes proposed by myself and the other reviewers. I am very happy to see how the paper has progressed and have one final minor spelling revisions:

- Line 32: Wrangel island is spelled Vrangel. It appears as Wrangel everywhere else.

Otherwise, I am satisfied with the paper and am looking forward to the publication!

Best,

Emil Karpinski

Reviewer #3:

Remarks to the Author:

The authors have done a good job of addressing the comments raised in my previous reviews.

My only remaining comment is that the authors should avoid using the term "significant" unless a statistical test has been performed.

RESPONSE TO THE EDITOR AND REVIEWERS

We would like to thank you for your consideration, valuable and friendly comments, and suggestions to improve this manuscript. Corrections are marked by **yellow** in manuscript and supplementary file. Queries are in black, our rebuttal in **blue**

REVIEWER COMMENTS

Reviewer #1 (Remarks to the Author):

I am happy with the authors responses to my final query.

Thank you for yours very important suggestions and contribution!

Reviewer #2 (Remarks to the Author):

I think the authors have done a great job incorporating the changes proposed by myself and the other reviewers. I am very happy to see how the paper has progressed and have one final minor spelling revisions:

- Line 32: Wrangel island is spelled Vrangel. It appears as Wrangel everywhere else.

Otherwise, I am satisfied with the paper and am looking forward to the publication!

Best,

Emil Karpinski

We corrected this misspelling.

Thank you, Emil! Your suggestions significantly improved our manuscript and got additional experience for us. Yours, Artem Nedoluzhko.

Reviewer #3 (Remarks to the Author):

The authors have done a good job of addressing the comments raised in my previous reviews.

My only remaining comment is that the authors should avoid using the term "significant" unless a statistical test has been performed.

Thank you for your contribution! We tried to avoid frequent use of “significant” in describing our results.